# C$^2$Prompt: Class-aware Client Knowledge Interaction for Federated Continual Learning

**Kunlun Xu**[*]
Wangxuan Institute of Computer Technology
Peking University
Beijing, China
`xkl@stu.pku.edu.cn`

**Yibo Feng**[*]
Wangxuan Institute of Computer Technology
Peking University
Beijing, China
`2022090917012@std.uestc.edu.cn`

**Jiangmeng Li**
University of Chinese Academy of Sciences
Beijing, China
`jiangmeng2019@iscas.ac.cn`

**Yongsheng Qi**
Inner Mongolia University of Technology
Hohhot, Inner Mongolia Autonomous Region
`qys@imut.edu.cn`

**Jiahuan Zhou** [†]
Wangxuan Institute of Computer Technology
Peking University
Beijing, China
`jiahuanzhou@pku.edu.cn`

## Abstract

Federated continual learning (FCL) tackles scenarios of learning from continuously emerging task data across distributed clients, where the key challenge lies in addressing both temporal forgetting over time and spatial forgetting simultaneously. Recently, prompt-based FCL methods have shown advanced performance through task-wise prompt communication. In this study, we underscore that the existing prompt-based FCL methods are prone to class-wise knowledge coherence between prompts across clients. The class-wise knowledge coherence includes two aspects: (1) intra-class distribution gap across clients, which degrades the learned semantics across prompts, (2) inter-prompt class-wise relevance, which highlights cross-class knowledge confusion. During prompt communication, insufficient class-wise coherence exacerbates knowledge conflicts among new prompts and induces interference with old prompts, intensifying both spatial and temporal forgetting. To address these issues, we propose a novel **C**lass-aware **C**lient Knowledge Interaction (**C$^2$Prompt**) method that explicitly enhances class-wise knowledge coherence during prompt communication. Specifically, a local class distribution compensation mechanism (LCDC) is introduced to reduce intra-class distribution disparities across clients, thereby reinforcing intra-class knowledge consistency. Additionally, a class-aware prompt aggregation scheme (CPA) is designed to alleviate inter-class knowledge confusion by selectively strengthening class-relevant knowledge aggregation. Extensive experiments on multiple FCL benchmarks demonstrate that C$^2$Prompt achieves state-of-the-art performance. Our source code is available at https://github.com/zhoujiahuan1991/NeurIPS2025-C2Prompt

---

[*]Equal contribution
[†]Corresponding author

# 1 Introduction

With the proliferation of edge computing and IoT devices [1], federated continual learning (FCL) has emerged as a critical paradigm for enabling intelligent systems to continuously learn from decentralized data streams while preserving data privacy [2; 3; 4; 5]. However, this setting presents a dual challenge: models must overcome catastrophic forgetting across sequential tasks (temporal dimension) while adapting to heterogeneous data distributions among clients (spatial dimension) [6; 7]. While traditional continual learning methods [8; 9; 10; 11; 12] and federated learning approaches [13; 14; 15; 16; 17] have made significant progress independently, their combined formulation in FCL struggles to address the superimposed forgetting effectively [6; 18; 19].

Existing FCL methods predominantly address the challenges of spatio-temporal knowledge transfer through data synthesis and parameter regularization [20]. However, data synthesis approaches [21; 22] typically depend on deep generative models trained on raw data, raising concerns regarding data privacy. In contrast, parameter regularization methods [3] attempt to balance learning and forgetting but often suffer from limited capacity to acquire new knowledge effectively. Recently, prompt-based learning [19; 23; 24] has emerged as a promising solution for FCL by maintaining task-specific prompts that store knowledge representations while leveraging a frozen pre-trained model [25; 26]. To overcome the overfitting to local distribution, some methods introduce inter-client prompt communication to

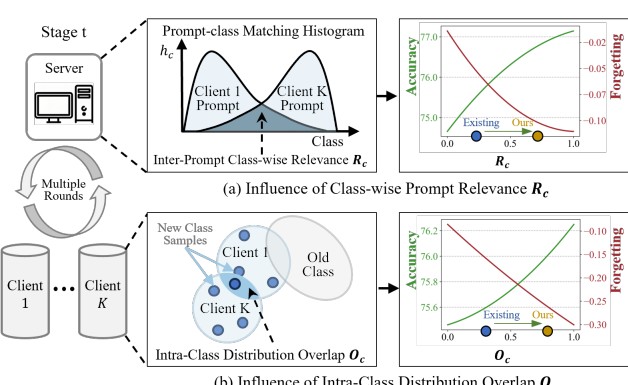

(a) Influence of Class-wise Prompt Relevance $R_c$

(b) Influence of Intra-Class Distribution Overlap $O_c$

Figure 1: In FCL, class-wise knowledge coherence includes two aspects: (a) inter-prompt class-wise relevance which influences prompt aggregation in the server, (b) intra-class distribution gap (overlap) across clients which influences the locally learned semantics of each class.

improve the robustness [19]. Despite their potential, these approaches are prone to class-wise knowledge coherence during prompt communication, which comprises two aspects.

First, as illustrated in Figure 1 (a), class-wise knowledge across prompts from different clients inherently varies to some extent. During prompt communication (aggregation), this divergence often results in knowledge conflicts, degrading the model's acquisition capacity. Moreover, these conflicts exacerbate forgetting, as the aggregated prompts may conflict with the historical prompts. Second, as depicted in Figure 1 (b), the intra-class distribution disparity across clients often affects the learned semantics of prompts. Certain features, although locally discriminative, may prove suboptimal from a global perspective, leading to further knowledge conflicts during prompt communication. Additionally, these locally discriminative features may be confused with historical data representations, resulting in degraded performance on previously learned tasks.

To address these challenges, we propose a novel Class-aware Client Knowledge Interaction ($C^2$Prompt) approach to improve inter-prompt class-wise relevance and intra-class distribution overlap simultaneously, as shown in Figure 1 (a)-(b). To achieve this, we first collect the local class distributions across clients and estimate the class-wise global distribution according to probability theory. Then, a local class distribution compensation mechanism (LCDC) is developed, which learns a set of class prompts to transfer the local semantics to the global domain, significantly improving intra-class knowledge consistency across clients. Additionally, each local prompt is recorded with its affinity with different classes. Then, a class-aware prompt aggregation scheme (CPA) is designed to exploit the class affinities to estimate the class-wise knowledge relevance across prompts and generate dynamic weights to enhance class-relevant knowledge aggregation, effectively alleviating the confusion caused by class knowledge conflict. Extensive experiments on multiple FCL benchmarks demonstrate that $C^2$Prompt outperforms state-of-the-art methods by large margins.

To summarize, the contributions of our paper are three-fold: (1) We present the $C^2$Prompt, an exemplar-free method that achieves Class-aware Client Knowledge Interaction to mitigate the tem-

poral and spatial forgetting simultaneously. (2) A local class distribution compensation mechanism is developed to complement local distribution to improve cross-client semantic consistency. (3) A class-aware prompt aggregation scheme is proposed to enhance intra-class knowledge aggregation and alleviate the knowledge conflict via a class-wise knowledge relevant estimation mechanism. (4) The superiority of $C^2$Prompt is validated on the challenging FCL benchmarks, where our method consistently achieves remarkable state-of-the-art performance.

# 2 Related Work

In this section, we review three research directions and discuss the state-of-the-art works that are most relevant to this paper.

## 2.1 Federated Learning

FL considers a distributed machine learning paradigm where decentralized data resources are modeled collaboratively [27; 28; 29; 30]. Each client trains with its corresponding data locally, and a server aggregates the client knowledge to obtain a global model [31; 32; 33; 34]. The key challenge in FL is the data heterogeneity problem, where the data are not independently and identically distributed (non-IID) on different clients [35; 36; 37; 38]. Current FL approaches can be primarily categorized into three branches, *i.e.*, client-side regularization, server-side regularization, and synthetic data generation [39]. Client-side regularization methods aim to improve the alignment with the global model by refining local updates [40; 41; 42; 43; 44; 45; 46; 47]. Server-side regularization approaches focus on achieving better aggregation to maximize the performance of the global model [48; 49; 50; 51]. Synthetic data generation methods rely on MixUp or training a deep generation model to generate synthetic data to approximate IID conditions [52; 53] or post-train the global model [54; 55; 56]. However, these FL methods assume that all the training data are available at the same time and neglect the practical condition that the training data occur sequentially.

## 2.2 Continual Learning

Continual Learning (CL) aims to learn with non-stationary data and generate a unified model that can address multiple tasks [9; 57; 58]. Current CL methods are mainly divided into two categories: rehearsal-based and rehearsal-free. Rehearsal-based methods [20; 21; 22] save a subset of learned samples into a memory buffer and replay them when learning a new task. While promising performance has been achieved, they usually require a large memory cost and raise privacy concerns during long-term learning. Rehearsal-free methods dynamically expand the network or isolate parameters for different tasks, regularize the network parameters that are important to learned tasks. Recently, freezing the pre-trained backbone model and only training a subset of learnable parameters is the current mainstream approach [59; 60; 61; 8]. L2P [62] pioneeringly introduced prompt learning to CL and proposed a key-query similarity method to select prompts for each task data from a prompt pool. CODAPrompt [11] transforms prompt selection into a differential process with an attention mechanism. However, these approaches only consider alleviating temporal forgetting and struggle to address the non-IID data in the federated learning scenario [63; 64; 65].

## 2.3 Federated Continual Learning

In FCL, each client continuously learns from a private and incremental task stream locally and a global model aims to aggregate the spatial-temporal knowledge in a unified model [3]. Existing FCL methods primarily focus on generative replay to address the spatial and temporal forgetting [66; 18; 6; 67; 68]. However, due to the slow convergence of generation training, training a generative model introduces massive training overheads [69]. Besides, the generative models typically risk privacy leakage of local information [6]. Recently, efficient tuning-based methods have shown advanced performance in FCL. PILoRA and LoRM introduced LoRA to address FCL by learning low-rank parameters in each client and aggregates them in the server. Besides, prompt learning has shown remarkable anti-forgetting capacity due to the parameter-matching mechanism that enables mutli-task knowledge co-consistency [70; 71; 19]. However, existing methods typically neglect the knowledge conflict between individual prompts during server-side aggregation, leading to significant knowledge loss.

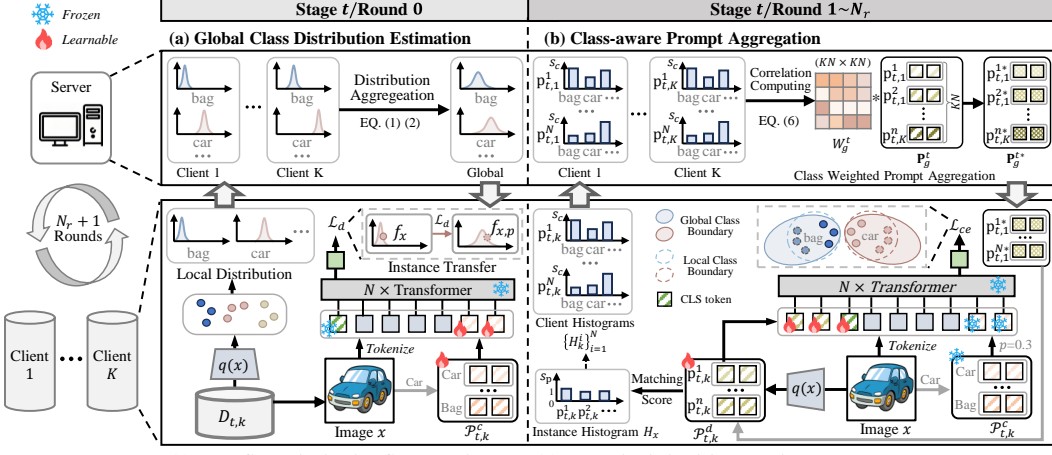

Figure 2: Overview of our C²Prompt approach. During the training stage $t$, given the data $D_{t,k}$ at each client $k$, the local class-aware feature distribution is collected and uploaded to the server to estimate the global distribution of each class. Then, the global distribution is distributed to the local clients to train client-specific class-distribution compensation prompts $\mathcal{P}_{t,k}^c$. During the later process, the discriminativity prompts $\mathcal{P}_{t,k}^d$ are introduced to learn classification-relevant knowledge, which are iteratively aggregated in the server according to the class knowledge relevance for $N_r$ rounds.

## 3 Preliminaries

**Problem Formulation:** In FCL, a collection of $K$ clients collaboratively learn under the coordination of a central server. Each client $k \in \{1, 2, \ldots, K\}$ sequentially learns a series of $T$ tasks. Let $\mathcal{T}_k^t$ denote the $t$-th task of the $k$-th client, and $D_k^t$ be its corresponding dataset. The model parameters of client $k$ during the learning of $\mathcal{T}_k^t$ are represented as $\theta_k^t$.

**Baseline:** Following recent FCL methods [19; 72; 71; 19], we adopt CODAPrompt [11] as the basic architecture for both clients and the server. For each local task $\mathcal{T}_k^t$, a set of prompts $\mathbf{P}_k^t \in \mathbb{R}^{N \times L_p \times D}$ is learned, where $N$ is the number of local prompts, $L_p$ is the length of each prompt, and $D$ is the input dimension of the Vision Transformer (ViT) encoder. On the server side, a global prompt pool $\mathbf{P}_g \in \mathbb{R}^{N_g \times L_p \times D}$ is maintained, containing prompts from both previous tasks $\mathcal{T}^{pre}$ and the current task $\mathcal{T}^{cur}$. The total number of prompts in the pool is denoted as $N_g = M \times N$, where $M$ is the number of seen tasks. For an image $\boldsymbol{x}$, its associated prompt $\mathbf{p}_x \in \mathbb{R}^{L_g \times D}$ is generated through a weighted sum of the prompts in $\mathbf{P}_g$:

$$\mathbf{p}_x = \sum_i^{N_g} \alpha_i [\mathbf{P}_g]_i \,, \tag{1}$$

where the weights $\boldsymbol{\alpha}_{\boldsymbol{x}} = \{\alpha_1, \alpha_2, \ldots, \alpha_{M_g}\}$ are computed based on query-key similarity:

$$\boldsymbol{\alpha}_{\boldsymbol{x}} = \{\gamma(q(x) \odot [\mathbf{A}_g]_1, [\mathbf{K}_g]_1), \gamma(q(x) \odot [\mathbf{A}_g]_2, [\mathbf{K}_g]_2), \ldots, \gamma(q(x) \odot [\mathbf{A}_g]_{N_g}, [\mathbf{K}_g]_{N_g})\}, \tag{2}$$

where $\gamma(\cdot, \cdot)$ represents the cosine similarity function, $\mathbf{K}_g \in \mathbb{R}^{M_g \times D}$ and $\mathbf{A}_g \in \mathbb{R}^{M_g \times D}$ are the learnable keys and attention weights of the prompts in $\mathbf{P}_g$. $\odot$ denotes the Hadamard product. For simplicity, given the one-to-one correspondence among $\mathbf{A}_g$, $\mathbf{K}_g$, and $\mathbf{P}_g$, we represent the global prompt pool as $\mathbf{P}_g$, encapsulating both attention and key representations.

In addition to the basic architecture of CODAPrompt, we also incorporate the knowledge distillation loss introduced by Powder [19] to enhance knowledge retention across tasks. The distillation loss is formulated as follows:

$$\mathcal{L}_{kd}(\hat{y}_{cu}, \hat{y}_{tr}) = -\sum_{k=0, k \neq y}^{K} [\hat{y}_{tr}]_k \log \frac{[\hat{y}_{cu}]_k}{[\hat{y}_{tr}]_k}, \tag{3}$$

where $\hat{y}_{cu}$ denotes the output logits of the current model, and $\hat{y}_{tr}$ represents the logits of the model at the beginning of the current communication round, containing the latest knowledge transferred from other tasks.

# 4 Proposed Method

In this section, we elaborate on our C$^2$Prompt which primarily consist of four modules, *i.e.*, Global Class Distribution Estimation, Class-aware Prompt Aggregation, Local Class Distribution Compensation, and Local Discriminativity Learning. An overview of C$^2$Prompt is illustrated in Figure 2, and the procedure is summarized in Algorithm 1 of Appendix B.

## 4.1 Global Distribution Generation

When the new stage data of a client $D_k^t$ is given, the local distribution for each class is first computed, resulting in a distribution set $\mathcal{D}_k^t = \{(\mu_{k,i}^t, \sigma_{k,i}^t)\}_{i=1}^{|\mathcal{C}_k^t|}$, where $(\mu_{k,i}^t, \sigma_{k,i}^t)$ denotes the class center and standard deviation, and $|\mathcal{C}_k^t|$ is the number of classes for client $k$. For each class $i$, the data distribution on client $k$ is approximated by a Gaussian distribution $\mathcal{N}(\mu_{i,k}^t, (\sigma_{i,k}^t)^2)$. Furthermore, the proportion of samples for class $i$ at client $k$ relative to the global sample size of class $i$ is represented as $p_{k,i}^t$. The global mean and standard deviation are then aggregated across all clients as follows:

$$\mu_i^g = \sum_{k=1}^K \mu_{i,k}^t p_{k,i}^t, \tag{4}$$

$$(\sigma_i^g)^2 = \sum_{k=1}^K \left((\mu_{i,k}^t)^2 + (\sigma_{i,k}^t)^2\right) p_{k,i}^t - (\mu_i^g)^2 \tag{5}$$

The **theoretical derivations** of Equation 4 and Equation 5 are provided in the Appendix A.1. Once calculated, the global distribution of each class is sent back to each client for further processing.

## 4.2 Local Class Distribution Compensation

Upon receiving the global class distribution, local class distribution compensation prompts $\mathcal{P}_{t,k}^c = \{\mathbf{p}_i^c\}_{i=1}^{|\mathcal{C}_k^t|}$ are introduced to address the issue of local undersampling by transferring the local samples to align with the global distribution. Specifically, for each class $i$, a local class distribution compensation prompt is denoted as $\mathbf{p}_i^c \in \mathbb{R}^{L_c \times d}$, where $L_c$ represents the length of $\mathbf{p}_i$. Given an input image $\boldsymbol{x}$, it is first tokenized [73] into a sequence representation $\boldsymbol{h_x} \in \mathbb{R}^{L_h \times d}$, where $L_h$ is the sequence length. The associated local class distribution compensation prompt of $\boldsymbol{x}$, denoted $\mathbf{p}_{\boldsymbol{x}}^c$, is obtained by indexing from $\mathcal{P}_{t,k}^c$ with its label. Both $\mathbf{p}_{\boldsymbol{x}}^c$ and $\boldsymbol{h_x}$ are then fed into transformer layers:

$$f_{x,p} = \boldsymbol{f}_\theta([\boldsymbol{h_x}, \mathbf{p}_{\boldsymbol{x}}^c, cls]), \tag{6}$$

where $\boldsymbol{f}_\theta$ is the parameters of the pretrained ViT, $cls$ is the [CLS] token [59], and $f_{x,p} \in \mathbb{R}^c$ is the generated feature. To ensure that $f_{x,p}$ aligns with the global class distribution, we assume that the global distribution for class $i$ follows a Gaussian parameterization $\mathcal{N}(\mu_i^g, (\sigma_i^g)^2)$. The alignment is enforced through a distribution-based cross-entropy loss that maximizes the likelihood of $f_{x,p}$ under the global distribution:

$$\mathcal{L}_c = -\frac{1}{2}(f_{x,p} - \boldsymbol{\mu}_i^g)^\top (\boldsymbol{\Sigma}_i^g)^{-1}(f_{x,p} - \boldsymbol{\mu}_i^g), \tag{7}$$

where $\boldsymbol{\Sigma}_i^g$ is a diagonal covariance matrix with its diagonal entries equal to $(\sigma_i^g)^2$. The **theoretical derivations** of Equation 7 are provided in the Appendix A.2. Note that once $\mathcal{P}_{t,k}^c = \{\mathbf{p}_i^c\}_{i=1}^{|\mathcal{C}_k^t|}$ is trained, it is frozen during the sequential rounds of training in the current stage.

### 4.3 Local Discriminativity Learning

When $\mathcal{P}_{t,k}^c$ is learned, we introduce the local discriminativity prompts $\mathcal{P}_{t,k}^d = \{\mathbf{p}_{t,k}^i\}_{i=1}^N$, matrixed as $\mathbf{P}_{t,k}^d$, which corresponds to prompts of original CODAPrompt, to enable new knowledge learning. Given $\boldsymbol{x}$ and its label $y$, we generate instance-specific discriminativity prompt $\mathbf{p}_{\boldsymbol{x}}^d$ form $\mathbf{P}_{t,k}^d$ according to Equation 1. Besides, the local class distribution compensation prompt $\mathbf{p}_{\boldsymbol{x}}^c$ is indexed from $\mathcal{P}_{t,k}^c$ using $y$. Then, a cross entropy loss ($CE$) is introduced to optimize $\mathbf{p}_{\boldsymbol{x}}^d$:

$$\mathcal{L}_{ce} = CE\big(\mathbf{W}_k \boldsymbol{f}_\theta([\boldsymbol{h}_{\boldsymbol{x}}, \mathbf{p}_{\boldsymbol{x}}^c, \mathbf{p}_{\boldsymbol{x}}^d, cls]), y\big), \tag{8}$$

where $\mathbf{W}_k$ is the learnable weight of the classifier of client $k$. Note that $\mathbf{p}_{\boldsymbol{x}}^c$ is exploited with $p = 0.5$ to sufficiently utilize the information of both local original data and the completed distributions.

At the same time, a instance histogram $H_{\boldsymbol{x}} = \{s_p^i\}_{i=1}^N$ for $\boldsymbol{x}$ is generated where $\{s_p^i\}$ denotes the similarity score between $\boldsymbol{x}$ and $\mathbf{p}_{t,k}^i$. During one round of training, we introduce a client histogram $H_k^i = \{s_c^j\}_{j=1}^{|\mathcal{C}_k^t|}$ for each prompt of stage $t$ that records the cumulative prompt-class matching scores $s_c^j$, which is mathematically calculated by:

$$s_c^j = \sum_{n=1}^{|D_{t,k}|} [H_{\boldsymbol{x}_n}]_j, \tag{9}$$

where $s_c^j$ represents the affinity between the prompt and class. Note that $H_k^i$ can be generated online during training and requires almost no additional computing overhead. When one round of discriminative prompt training is finished, a set of client histograms $\{H_k^i\}_{i=1}^N$ for the new stage prompts is uploaded to the server.

### 4.4 Class-aware Prompt Aggregation

When a round of local discriminativity learning is finished, the local client histograms are collected to form a set $\mathcal{H}_g^t = \{H_1^i\}_{i=1}^N \cup \{H_2^i\}_{i=1}^N \cup \cdots \cup \{H_K^i\}_{i=1}^N$, which is matrixed as $\mathbf{H}_g^t \in \mathbb{R}^{KN \times |\mathcal{C}_t|}$. Then, an inter-prompt correlation matrix $W_g^t \in \mathbb{R}^{KN \times KN}$ is computed by

$$W_g^t = \gamma(\mathbf{H}_g^t {\mathbf{H}_g^t}^\top / \tau), \tag{10}$$

where $\gamma$ is the softmax function that is conducted row-wise here, and $\tau$ is a hyperparameter to scale the similarity scores. Besides, the prompts of stage $t$ are also collected from the clients to form a set $\mathcal{P}_g^t = \{p_1^i\}_{i=1}^N \cup \{p_2^i\}_{i=1}^N \cup \cdots \{p_K^i\}_{i=1}^N$, which is matrixed as $\mathbf{P}_g^t \in \mathbb{R}^{KN \times L_p \times d}$. Then, a Class Weighted Prompt Aggregation process is conducted by:

$$\mathbf{P}_g^{t*} = W_g^t \mathbf{P}_g^t, \tag{11}$$

where $\mathbf{P}_g^{t*} \in \mathbb{R}^{KN \times L_p \times d}$ is the updated prompts that have collected the most relevant knowledge from prompts of different clients. Then, $\mathbf{P}_g^{t*}$ is split into $K$ prompt sets and distributed to the corresponding clients.

**Training and Inference:** As shown in Figure 2, during stage $t$, the training process consists of two phases. Firstly, Global Class Distribution Estimation and Local Class Distribution Compensation are conducted at round 0. The local distribution compensation loss $\mathcal{L}_c$ is adopted to train $\mathcal{P}_{t,k}^c$. Then, from round 1 to $N_r$, Class-aware Prompt Aggregation and Local Discriminativity Learning are conducted in turn. The model is optimized by an overall loss:

$$\mathcal{L}_d = \mathcal{L}_{ce} + \beta \mathcal{L}_{kd}, \tag{12}$$

where $\beta$ is a hyperparameter to balance the loss components.

During inference, following previous works [19], the prompts learned on all the seen local tasks are collected to generate a prompt $\mathbf{p}_{\boldsymbol{x}}$ which is exploited to generate predictions by

$$\hat{y} = \gamma\big(\mathbf{W}_g \boldsymbol{f}_\theta([\boldsymbol{h_x}, \boldsymbol{p_x}, cls]), y\big), \tag{13}$$

where $\mathbf{W}_g$ is the global classifier by concentrates the local classifiers learned from different tasks following [19].

## 5 Experiments

### 5.1 Experimental Setups

**Datasets and Metrics:** We conduct the experiments on three widely used benchmarks in FCL, *i.e.*, ImageNet-R[74], DomainNet[75] and CIFAR-100[76]. To evaluate the effectiveness of different FCL methods, 6 metrics are adopted in this paper, including Average Accuracy (Avg), Average Incremental Accuracy (AIA), Forgetting Measure (FM), Forward Transfer (FT), Backward Transfer (BT), Combined Transfer (CT). The configurations of the benchmarks and the details of the metrics are presented in Appendix C.

Table 1: Result comparison on the ImageNet-R and DomainNet benchmark

| Methods | Pub. | ImageNet-R | | | | | | DomainNet | | | | | |
|---|---|---|---|---|---|---|---|---|---|---|---|---|---|
| | | Avg↑ | AIA↑ | FM↓ | FT↑ | BT↑ | CT↑ | Avg↑ | AIA↑ | FM↓ | FT↑ | BT↑ | CT↑ |
| FedWEIT | *ICML2021* | 71.10 | 74.30 | 1.80 | -2.39 | -1.83 | -3.86 | 67.84 | 69.63 | 1.91 | -2.92 | -3.11 | -4.97 |
| CFeD | *IJCAI2022* | 47.93 | 59.79 | 3.81 | -17.67 | -14.92 | -29.60 | 42.85 | 60.19 | 1.65 | -4.98 | -13.32 | -15.64 |
| GLFC | *CVPR 2022* | 72.96 | 75.21 | 1.10 | -3.87 | -1.55 | -5.11 | 69.75 | 70.34 | 1.23 | -4.08 | -2.46 | -6.04 |
| Fedspace | *CVPR 2023* | 72.27 | 73.36 | 2.01 | -2.60 | -4.91 | -5.95 | 68.98 | 70.71 | 1.80 | 1.87 | -4.16 | -1.45 |
| Fed-L2P | *CVPR2022* | 77.88 | 75.03 | 0.41 | -2.79 | -0.17 | -2.92 | 70.98 | 72.36 | 0.16 | -2.18 | 0.10 | -2.09 |
| Fed-Dual | *ECCV2022* | 76.85 | 74.91 | 0.49 | -3.12 | 0.22 | -2.95 | 71.90 | 72.15 | 0.16 | -1.82 | 0.41 | -1.49 |
| Fed-CODA | *CVPR2023* | 79.65 | 75.14 | **-0.68** | -2.53 | 1.69 | -1.18 | 72.47 | 72.84 | 0.01 | -0.82 | **0.83** | -0.15 |
| Fed-CP | *ICML2023* | 76.75 | 72.59 | 0.63 | -3.16 | 0.00 | -3.16 | 71.28 | 69.92 | 0.18 | -2.78 | 0.00 | -2.78 |
| Powder | *ICML2024* | 84.69 | 84.08 | -0.54 | 4.48 | **1.95** | 6.04 | 75.98 | 77.28 | 0.10 | 1.28 | 0.14 | 1.40 |
| PILoRA | *ECCV2024* | 45.43 | 48.72 | 0.92 | -5.75 | -7.32 | -12.54 | 31.22 | 40.76 | 0.55 | -0.12 | -0.74 | -1.81 |
| Fed-MOS | *AAAI2025* | 47.67 | 47.08 | 1.40 | -3.30 | -0.12 | -3.37 | 40.37 | 45.22 | 0.31 | -1.43 | -1.21 | -2.50 |
| LoRM | *ICLR2025* | 58.00 | 67.78 | 8.71 | -4.67 | -9.22 | -13.70 | 23.18 | 28.49 | 5.72 | -1.32 | -0.11 | -1.40 |
| C²Prompt | *This Paper* | **87.20** | **85.93** | -0.36 | **7.63** | 1.12 | **8.52** | **78.88** | **77.55** | **-0.02** | **3.87** | 0.23 | **4.05** |

**Compared Methods:** We compare our proposed C²Prompt with the following methods: (1) Fully-Tuning-based (FULLY) federated continual learning methods, including FedWEIT [77], CFeD [78], GLFC [7] and FedSpace [79]. (2) Efficient-Tuning-based (EFFICIENT) methods, including prompt learning approaches, FedCPrompt [71] and Powder [19]. Besides, the state-of-the-art prompt-based continual learning methods, *i.e.*, L2P [62], DualPrompt [8], CODAPrompt [11], are integrated with the well-known FedAvg [48] algorithm to make a comprehensive comparison (Fed-L2P, L2P-Dual, Fed-CODAP, Fed-CPrompt). Additionally, the LoRA-based FCL methods, including PILoRA [80] and LoRM [81], and the adapter-based continual learning method MOS [82] is integrated with FedAvg to form Fed-MOS. All experiments are implemented using official code, with the ViT-B/16 pre-trained on ImageNet-21k serving as the backbone network.

**Implementation Details** The settings of our discriminativity prompts follow the configuration of previous works [19], where $L_p$, $N$ and $d$ are set to 10, 8 and 768, respectively. For our class distribution compensation prompt, the prompt length $L_c$ is set to 3 by default. The Adam optimizer with a learning rate of 0.01 is adopted during training. For all the experiments, the training and testing images are resized to 224×224. The client number $K$ and round number for each task are set to 5 and 3, respectively. All experiments are conducted on a single Nvidia 4090 GPU.

### 5.2 Comparison Results

We follow the experimental setting of the previous methods [19] and the comparison results on the ImageNet-R and DomainNet are represented in Table 1, where Avg and AIA are the most important

metrics indicating the long-term knowledge accumulation and progressive performance, respectively. The best and second best methods are highlighted in **Bold** and Underlined, separately.

**Avg Comparison:** Our C$^2$Prompt outperforms the state-of-the-art Powder, achieving improvements of **2.51%** and **2.90%** on ImageNet-R and DomainNet, respectively. These results demonstrate the superiority of our method in long-term knowledge consolidation. This is because the new knowledge acquisition capacity is significantly improved with our local class distribution compensation and class-aware discriminativity prompt aggregation designs. Besides, the accurate knowledge communication mechanism avoids the irrelevant prompts fusion that generate invalid prompts which not only semantically away from new prompts, but also conflict with historical prompts.

**AIA Comparison:** Our C$^2$Prompt achieves improvements of **1.85%** on ImageNet-R and also outperforms all existing approaches on DomainNet, verifying our method consistently obtains superior performance compared the existing methods across different training stages. This is attributed to the local class distribution compensation and class-aware discriminativity prompt aggregation designs that enhance robust local knowledge acquisition and improve distributed knowledge collection.

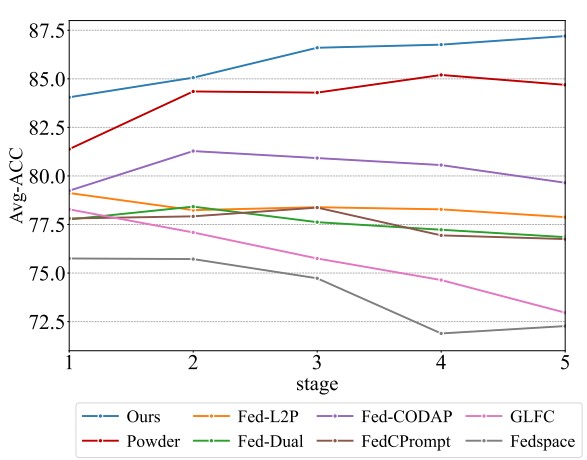

Figure 3: Avg-ACC curves on the seen tasks across training stages .

**FM Comparison:** Fed-CODAP, Powder and our C$^2$Prompt show a negative forgetting rate on the small-scale dataset ImageNet-R. This indicates the new tasks can facilitate historical task learning when training samples are limited. On the large-scale dataset DomainNet, only our C$^2$Prompt shows a negative forgetting rate. These results verify the effective antiforgetting capacity of our method under different conditions.

**FT Comparison:** Our C$^2$Prompt shows advanced forward-transfer capacity, outperforming existing methods by at least **3.15%** and **2.59%** on ImageNet-R and DomainNet, respectively. This is primarily attributed to the Global Class Distribution Estimation and Local Class Distribution Compensation designs, where the former can effectively exploit the asynchronously arriving data of the same class to generate reliable global distribution estimation. Then the estimated global distribution is exploited by the later to achieve data-level information compensation, thereby significantly improving subsequent data learning.

**BT Comparison:** Fed-CODAP, Powder and our C$^2$Prompt consistently show positive backward-transfer across both ImageNet-R and DomainNet datasets. This is because the asynchronously arriving data enable the later tasks to enhance the knowledge of previously seen classes. We observe that the backward-transfer results of C$^2$Prompt are relatively inferior to Fed-CODAP and Powder. This is because the Local Class Distribution Compensation and Class-aware Prompt Aggregation de-

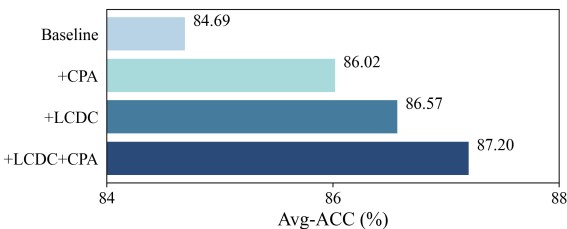

Figure 4: Ablation on the model components.

signs significantly improve the distributed data learning capacity at each stage, leaving less improvement space for seen tasks.

**CT Comparison:** As for the combined transfer of forward and backward, our C$^2$Prompt outperforms all existing approaches with **2.48%** and **2.65%** improvements on ImageNet-R and DomainNet, respectively. These results demonstrate that the class-aware client knowledge interaction designs in this paper effectively boost the overall learning capacity in the temporal dimension in FCL. Specifically,

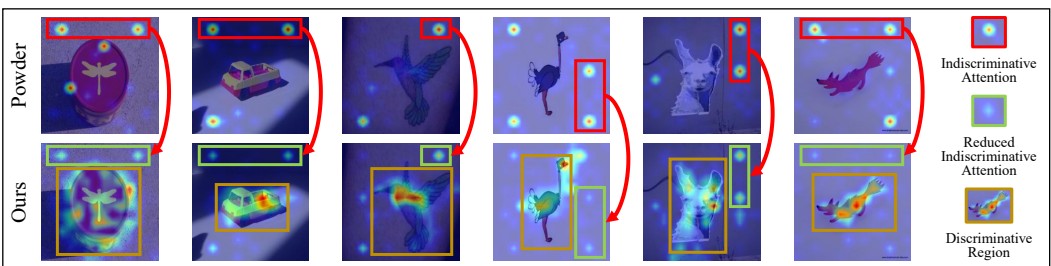

Figure 5: Visualization of attention across prompt and image regions.

the Global Class Distribution Estimation effectively aggregates the distributional information across spatial and temporal data sources. Local Class Distribution Compensation module leverages the global distributional image to overcome the non-IID phenomenon across clients. Finally, Local Discriminativity Learning and Class-aware Prompt Aggregation modules effectively integrate the distributional knowledge into the prompts.

**Performance Tendency Analysis:** To further analyze the model learning process, we visualize the Average Accuracy (Avg-ACC) across the seen tasks during the FCL stages in Fig. 3 on the ImageNet-R benchmark. The results show that our method consistently outperforms state-of-the-art approaches across all stages. Furthermore, the Avg-ACC of $C^2$-Prompt exhibits a stable upward trajectory throughout the training process, whereas other methods display either declining or fluctuating trends. This advantage is attributed to our class-aware client knowledge interaction designs, which effectively extract and preserve robust knowledge over long-term training. In contrast, existing methods are more prone to knowledge conflicts during parameter aggregation in FCL, leading to performance degradation as training progresses.

## 5.3 Ablation Study and Additional Analysis

**Ablation on components**. Since Fig. 2 (a) and Fig. 2 (c) rely on each other, we present them as a unified component termed LCDC. Besides, Fig. 2 (b) and Fig. 2 (d) also rely on each other, we present them as a unified component termed CPA.

The ablation studies on LCDC and CPA are illustrated in Fig. 4, which are conducted with the ImageNet-R benchmark. When using LCDC module alone, our method obtains **1.88%** improvement compared to the baseline, verifying the effectiveness of the Global Class Distribution Estimation and Local Class Distribution Compensation mechanism. Besides, CPA achieves

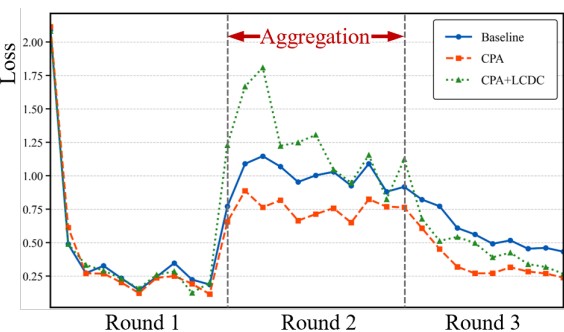

Figure 6: Visualization of loss curves.

**1.33%** improvement compared to the baseline, demonstrating the effectiveness of our Class-aware Prompt Aggregation design. When all our modules are used together, the model performance is further improved with **2.51%** improvement. This is because LCDC and CPA achieve input-level class information compensation and feature extraction parameter-level knowledge communication, respectively. Therefore, they are complementary to each other.

We also visualize the loss of Baseline, CPA and CPA+LCDC in Figure 6. During round 1, different methods primarily learn with new data and converge similarly. When the first aggregation is conducted, all the methods show improved loss due to the parameter drift. CPA shows the least loss improvement due to the class-aware prompt fusion design that mitigates the knowledge conflict issue. CPA+LCDC shows a large loss improvement because the class distribution Compensation design guides the model in the early stage. During the second aggregation, the loss improvement of CPA+LCDC is significantly reduced since knowledge correlation between prompts is improved after round 2. After the training of round 3, both CPA and CPA+LCDC show significantly lower loss compared to the

Baseline. Although CPA and CPA+LCDC obtain similar final losses, the performance of CPA+LCDC is superior to CPA since LCDC improves the robustness of the learned knowledge.

**Visualization of learned prompts:** Figure 5 illustrates the prompt attention maps of our $C^2$-Prompt in comparison with state-of-the-art Powder [19]. Specifically, the prompts generated by Powder are largely dominated by class-irrelevant knowledge and exhibit limited discriminative feature extraction capacity. In contrast, the prompts generated by our method effectively focus on the discriminative regions and influence less on the class-irrelevant knowledge. These improvements are primarily attributed to our Class-aware Prompt Aggregation mechanism, which effectively alleviates conflicted knowledge fusion during prompt aggregation.

**Communication Overhead:** Table 2 compares the communication and parameter overhead of $C^2$-Prompt with state-of-the-art methods. Our approach demonstrates comparable communication and parameter costs with Powder, with only 0.6% and 6.8% increases, respectively. The additional communication overhead stems from the exchange of class distribution information between the server and clients. However, due to the sparse distribution of classes, this overhead remains minimal. The slight increase in training parameter count is attributed to the introduction of local class distribution compensation prompts, which are significantly fewer than the discriminative prompts commonly used in existing methods. Note that $C^2$-Prompt **does not** introduce any additional parameters or computational overhead during inference, as only the discriminative prompts are employed for testing.

Table 2: Comparison of communication and additional parameter overhead.

| Methods | Communication | Parameter | |
| --- | --- | --- | --- |
| | | Training | Inference |
| Fed-L2P | 686.69MB | 3.96MB | 3.96MB |
| Fed-Dual | 621.78MB | 4.73MB | 4.73MB |
| Fed-CODAP | 815.63MB | 11.43MB | 11.43MB |
| Fed-CPrompt | 815.63MB | 11.43MB | 11.43M |
| Powder | **493.08MB** | **2.64MB** | **2.64MB** |
| $C^2$Prompt | 496.01MB | 2.82MB | **2.64MB** |

## 6 Conclusion

In this paper, we propose Class-aware Client Knowledge Interaction ($C^2$Prompt), which enhances the class-wise knowledge coherence between prompts across clients, significantly alleviating both temporal and spatial forgetting by mitigating the potential knowledge conflict during prompt communication. $C^2$Prompt introduces two kinds of prompts, local class distribution compensation prompt and local discriminativity prompt. The former transfers local class features to a global class-wise distribution to improve the intra-class semantic consistency across clients. The latter learn discrimination capacity with local data and aggregated with the ones from other clients in the server according to the class-wise affinity, enabling class-wise knowledge enhancement while alleviating conflicts. Extensive experiments on the challenging FCL benchmarks demonstrate that our method significantly outperforms the state-of-the-art, validating the effectiveness of our approach.

**Limitation Discussion:** Our approach requires class distribution communication at the initial round. Although this operation incurs minimal overhead due to the sparsity of distributional parameters, it introduces a minor communication cost. Furthermore, the prompt aggregation process generates client-specific prompts, slightly increasing computing and storage overhead compared to existing prompt-based methods. Nevertheless, this remains significantly more efficient than full fine-tuning approaches, as the number of learnable parameters in prompts is substantially smaller than that of the entire feature extractor.

## Acknowledgement

This work was supported by the National Key R&D Program of China (2024YFA1410000), the National Natural Science Foundation of China (62376011), and the China Postdoctoral Science Foundation (2025T180417).

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

# A  Theoretical Justification of Distribution Operations

## A.1  Server-side Distribution Aggregation

Federated continual learning (FCL) involves decentralized clients collaboratively learning over sequential tasks. To enhance cross-client coherence, we estimate the global class distributions on the server by aggregating local class distributions from each client. Given local class distributions $\mathcal{N}(\mu_1, \sigma_1^2), \mathcal{N}(\mu_2, \sigma_2^2), \ldots, \mathcal{N}(\mu_n, \sigma_n^2)$ with probability density functions $f_1(x), f_2(x), \ldots, f_n(x)$, the global distribution is defined using the clients' sample frequencies $p_1, p_2, \ldots, p_n$, satisfying $\sum_{i=1}^{n} p_i = 1$.

The global class mean is calculated as:

$$\mu = \int \left[ p_1 f_1(x) + p_2 f_2(x) + \cdots + p_n f_n(x) \right] x \, dx = p_1 \mu_1 + p_2 \mu_2 + \cdots + p_n \mu_n \tag{14}$$

The global class variance is expressed as:

$$\sigma^2 = \int \left[ p_1 f_1(x) + p_2 f_2(x) + \cdots + p_n f_n(x) \right] x^2 \, dx - \mu^2 \tag{15}$$

Then we have

$$\sigma^2 = p_1 \left( \sigma_1^2 + \mu_1^2 \right) + p_2 \left( \sigma_2^2 + \mu_2^2 \right) + \cdots + p_n \left( \sigma_n^2 + \mu_n^2 \right) - \mu^2 \tag{16}$$

This aggregation provides a comprehensive global distribution for each class, continuously updated as new tasks arrive. These global statistics are then communicated back to clients to guide local prompt optimization, enhancing semantic consistency.

## A.2  Local Class Distribution Compensation Loss

Upon receiving global class distributions, each client optimizes local class distribution compensation prompts. The derivation of the loss function for the prompt is as follows:

For class $i$, the global class distribution is represented as $\mathcal{N}(\mu_g^i, \Sigma_g^i)$. Specifically, for feature vector $X \in \mathbb{R}^d$ of class $i$, its probability density is:

$$p(X \mid \mu_i, \Sigma_i) = \frac{1}{(2\pi)^{d/2} |\Sigma_i|^{1/2}} \cdot e^{-\frac{1}{2}(X - \mu_i)^\top \Sigma_i^{-1}(X - \mu_i)} \tag{17}$$

where $\mu_i \in \mathbb{R}^d$ is the mean vector and $\Sigma_i \in \mathbb{R}^{d \times d}$ is a positive definite symmetric covariance matrix. When assuming independent feature dimensions, the covariance matrix reduces to diagonal form:

$$\Sigma_i = \mathrm{diag}(\sigma_{i1}^2, \sigma_{i2}^2, \ldots, \sigma_{id}^2) \tag{18}$$

with determinant and inverse matrix given by:

$$|\Sigma_i| = \prod_{j=1}^{d} \sigma_{ij}^2 \quad \text{and} \quad \Sigma_i^{-1} = \mathrm{diag}\left( \frac{1}{\sigma_{i1}^2}, \frac{1}{\sigma_{i2}^2}, \ldots, \frac{1}{\sigma_{id}^2} \right) \tag{19}$$

According to (17), the exponent expands to:

$$(X - \mu_i)^\top \Sigma_i^{-1} (X - \mu_i) = \sum_{j=1}^{d} \frac{(X_j - \mu_{ij})^2}{\sigma_{ij}^2} \tag{20}$$

Thus, the probability density function decomposes as:

$$p_i(X | \mu_i, \sigma_i^2) = \frac{1}{(2\pi)^{d/2} \left( \prod_{j=1}^{d} \sigma_{ij}^2 \right)^{1/2}} \cdot e^{-\frac{1}{2} \sum_{j=1}^{d} \frac{(X_j - \mu_{ij})^2}{\sigma_{ij}^2}} \tag{21}$$

In federated learning, client-generated features $f_{x,p}$ should align with the server's global class distribution $\mathcal{N}(\mu_g^i, \Sigma_g^i)$. The optimization objective becomes maximizing the log-likelihood:

$$\mathcal{L}_c = \log p(f_{x,p}|\mu_g^i, \Sigma_g^i) \tag{22}$$

Substituting (17) and expanding:

$$\log p(f_{x,p}|\mu_g^i, \Sigma_g^i) = -\frac{d}{2}\log(2\pi) - \frac{1}{2}\log|\Sigma_g^i| - \frac{1}{2}(f_{x,p} - \mu_g^i)^\top(\Sigma_g^i)^{-1}(f_{x,p} - \mu_g^i) \tag{23}$$

Minimizing the negative log-likelihood loss:

$$\mathcal{L}_c = -\log p(f_{x,p}|\mu_g^i, \Sigma_g^i) = \frac{d}{2}\log(2\pi) + \frac{1}{2}\log|\Sigma_g^i| + \frac{1}{2}(f_{x,p} - \mu_g^i)^\top(\Sigma_g^i)^{-1}(f_{x,p} - \mu_g^i) \tag{24}$$

Since the first two constant terms can be omitted for optimization, $\mathcal{L}_c$ can be simplified to:

$$\mathcal{L}_c = \frac{1}{2}(f_{x,p} - \mu_g^i)^\top(\Sigma_g^i)^{-1}(f_{x,p} - \mu_g^i) \tag{25}$$

This objective encourages local features to match the global class distribution, effectively reducing inter-client distribution gaps and enhancing semantic coherence during federated updates.

### A.3 Theoretical Implications

Overall, the proposed distribution operations theoretically guarantee smoother knowledge alignment across clients by optimizing class-wise distribution coherence. This process stabilizes global knowledge representations throughout continual learning, improving the robustness of the learned knowledge and improving the aggregation compatibility between prompts from different clients, effectively mitigating spatial and temporal forgetting.

## B  Algorithm of the proposed approach

The overall process of our $C^2$Prompt is shown in Algorithm 1.

## C  Details of the datasets and evaluation metrics

### C.1  Datasets

We use three image datasets commonly utilized in Federated Continual Learning (FCL) to evaluate our method: ImageNet-R, DomainNet, and CIFAR-100. ImageNet-R consists of 30,000 images from 200 categories, including challenging samples from ImageNet and newly collected samples with various styles. The dataset is divided into a training set with 24,000 images and a test set with 6,000 images, and 20% of the training set is selected as a validation set for tuning model parameters. DomainNet is a large dataset containing 600,000 images and 345 categories, spanning six different domains. CIFAR-100 contains 50,000 training and 10,000 test-colored images for 100 classes, respectively.

### C.2  Configuration of Federated Continual Learning Benchmarks

The benchmark configuration of this paper follows previous FCL method Powder [19]. Based on transferability (tasks have class overlap and each task contains only a small portion of each class's data) and asynchrony, for ImageNet-R, each task randomly selects 20 classes (20% samples per class), distributed randomly across clients with varying round durations. For DomainNet, each task randomly selects 35 classes (2% samples per class due to closeness to pre-trained distribution, others same as ImageNet-R). We control task overlap by randomly selecting classes with the least overlap to

---

**Algorithm 1** Local Class Distribution Compensation and Global Class Distribution Estimation

---

**Input:** Stage $t$ data $\mathcal{D}^t = \{\mathcal{D}_k^t\}_{k=1}^K$

**Output:** Global prompt pool $\mathbf{P}_g^t$

Initialize $\mathcal{P}_d^{t,k*} = None$
**for** each round $r = 0$ to $N_r$ **do**
   **if** round r = 0 **then**

      *# Local Class Distribution Compensation ($\boldsymbol{LCDC}$)*
      **for** each client $k$ **do**
         Compute local class statistics $(\mu_{k,i}^t, \sigma_{k,i}^t)$ for each class $i$
         Upload $(\mu_{k,i}^t, \sigma_{k,i}^t)$ to the server
      **end for**

      *# Global Class Distribution Estimation*
      Estimate global class center $\mu_i^g = \sum_{k=1}^K \mu_{i,k}^t p_{k,i}^t$, Eq. 4

      Estimate global class variance $(\sigma_i^g)^2 = \sum_{k=1}^K \left( (\mu_{i,k}^t)^2 + (\sigma_{i,k}^t)^2 \right) p_{k,i}^t - (\mu_i^g)^2$, Eq. 5

      Distribute the global distribution to the corresponding clients

      *# Back to ($\boldsymbol{LCDC}$)*
      **for** each client $k = 1$ to $K$ **do**
         Initialize local class distribution complension prompts $\mathcal{P}_{t,k}^c = \{p_i^c\}_{i=1}^{|C_t^k|}$
         For input $x$, obtain $f_{x,p} = f_\theta([\boldsymbol{h}_x, \mathbf{p}_{\boldsymbol{x}}^c, [\text{CLS}]])$, Eq. 6
         Update $\mathcal{P}_{t,k}^c$ using $\mathcal{L}_c = -\frac{1}{2}(f_{x,p} - \boldsymbol{\mu}_i^g)^\top (\boldsymbol{\Sigma}_i^g)^{-1}(f_{x,p} - \boldsymbol{\mu}_i^g)$, Eq. 7
      **end for**
      Froze prompt $\mathcal{P}_{t,k}^c$
   **end if**

   *# Local Discriminativity Learning*
   **for** each client $k = 1$ to $K$ **do**
      **if** $\mathcal{P}_{t,k}^{d*} \neq None$ **then**
         Initialize local discriminativity prompts $\mathcal{P}_{t,k}^d = \{p_i^d\}_{i=1}^N$ with $\mathcal{P}_{t,k}^{d*}$
      **end if**
      For each input $x$, pbtain $\mathbf{p}_{\boldsymbol{x}}^d, \mathbf{p}_{\boldsymbol{x}}^c$ and $H_{\boldsymbol{x}}$
      Update using $\mathcal{L}_{ce} = CE\big(\mathbf{W}_k \boldsymbol{f}_\theta([\boldsymbol{h}_x, \mathbf{p}_{\boldsymbol{x}}^c, \mathbf{p}_{\boldsymbol{x}}^d, cls]), y\big)$, Eq. 8
      Obtain client histogram $H_k^i = \{s_c^j\}_{j=1}^{|\mathcal{C}_k^t|}$ for prompt $p_d^i$, where $s_c^j = \sum_{n=1}^{|D_{t,k}|}[H_{\boldsymbol{x}_n}]_j$, Eq. 9
      Upload $\{H_k^i\}_{i=1}^N$ and $\mathcal{P}_{t,k}^d$ to the server
   **end for**

   *# Class-aware Prompt Aggregation ($\boldsymbol{CPA}$)*
   Server collects all client histograms to form $\mathbf{H}_g^t \in \mathbb{R}^{KN \times |\mathcal{C}_t|}$
   Server collects all client discriminativity prompts to form $\mathbf{P}_g^t$
   Compute inter-prompt attention: $W_g^t = \gamma(\mathbf{H}_g^t \mathbf{H}_g^{t\top}/\tau)$, Eq. 10
   Update prompts: $\mathbf{P}_g^{t*} = W_g^t \mathbf{P}_g^t$, Eq. 11
   Distribute $\mathcal{P}_{t,k*}^d$ to corresponding clients
**end for**
Return $\mathbf{P}_g^t$

---

study FCL performance under different task correlations. Unlike the common Dirichlet distribution method in FL, we avoid it here because in FCL, class sets of different tasks vary greatly, making it hard to control similarity with it. The setup details for the CIFAR-100 dataset are the same as

those for ImageNet-R. Simultaneously, we set five clients for training, each executing distinct tasks. Furthermore, we organize the training process into phases, each consisting of three communication rounds. At the beginning of each phase, 40% of the clients are selected to initiate learning on new tasks. To ensure the fairness of the results, we keep the optimizer, learning rate, and local training epochs consistent with those of the Powder method when training the classification prompts. Our LCDC is trained using the Adam optimizer when new tasks arrive, and the trained prompt is only used for the training of the formal classification prompt and not for the testing phase.

## C.3 Evaluation Metrics

We evaluate the effectiveness of our method by adapting seven metrics, including the Average accuracy of all tasks (Avg), Average Incremental Accuracy (AIA)[83], Forgetting Measure (FM)[84], Forward Transfer (FT)[85], Backward Transfer (BT)[85], Combined Transfer (CT), Final Average Accuracy (FAA).

**Average accuracy of all tasks (Avg)**     This metric measures the average accuracy of the final model across all tasks, computed as

$$\text{Avg} = \frac{1}{|\mathcal{T}|} \sum_{\mathcal{T}_c^t \in \mathcal{T}} a_{c,\max(\mathcal{R})}^t$$

where $\mathcal{T}$ denotes the set of all tasks during the Federated Continual Learning (FCL) process, and $a_{c,\max(\mathcal{R})}^t$ denotes the final accuracy of task $\mathcal{T}_c^t$ (i.e., the accuracy on this task when training concludes).

**Average Incremental Accuracy (AIA)**     This metric measures the average accuracy over the FCL process, computed as

$$\text{AIA} = \frac{1}{|\mathcal{R}|} \sum_{r \in \mathcal{R}} \sum_{\mathcal{T}_c^t \in \mathcal{T}_r} a_{c,r}^t$$

where $\mathcal{R}$ denotes the set of rounds with task switch, $\mathcal{T}_r$ denotes the set of existing tasks at round $r$, and $a_{c,r}^t$ denotes the accuracy of $\mathcal{T}_c^t$ at round $r$.

**Forgetting Measure (FM)**     Forgetting is measured by the difference between the highest historical accuracy and the current accuracy of a task. This metric quantifies the model's memory stability by the average forgetting over the FCL process, computed as

$$\text{FM} = \frac{1}{|\mathcal{R}|} \sum_{r \in \mathcal{R}} \left( \sum_{\mathcal{T}_c^t \in \mathcal{T}_r} a_{c,r}^t - \tilde{a}_{c,r}^t \right)$$

where $\tilde{a}_{c,r}$ denotes the max accuracy of $\mathcal{T}_c^t$ before round $r$.

**Forward Transfer (FT)**     This metric assesses the model's ability to transfer knowledge into a task, from both previously learned tasks and other currently learned tasks, computed as

$$\text{FT} = \frac{1}{|\mathcal{T}|} \sum_{\mathcal{T}_c^t \in \mathcal{T}} \left( \acute{a}_c^t - \hat{a}_c^t \right)$$

where $\mathcal{T}$ denotes all tasks during the FCL process, $\acute{a}_c^t$ denotes the accuracy of $\mathcal{T}_c^t$ when it finished, and $\hat{a}_c^t$ denotes the accuracy of single-task training.

**Backward Transfer (BT)**     This metric evaluates the model's ability to transfer knowledge from new tasks back to previously learned tasks, computed as

$$\text{BT} = \frac{1}{|\mathcal{T}|} \sum_{\mathcal{T}_c^t \in \mathcal{T}} \left( a_{c,\max(\mathcal{R})}^t - \acute{a}_c^t \right)$$

where $a_{c,\max(\mathcal{R})}^t$ denotes the final accuracy of $\mathcal{T}_c^t$.

**Combined Transfer (CT)**   This metric is a combination of FT and BT, evaluating the amount of information that a task $\mathcal{T}_c^t$ acquires from other tasks. The other tasks can have any sequence relationship with task $\mathcal{T}_c^t$ in terms of temporal dimension. It is computed as

$$\mathrm{CT} = \frac{1}{|\mathcal{T}|} \sum_{\mathcal{T}_c^t \in \mathcal{T}} \left( a_{c,\max(\mathcal{R})}^t - \hat{a}_c^t \right)$$

**Final Average Accuracy (FAA)**   FAA is a standard metric used in FCIL to measure knowledge retention and accumulation. Let $a^t$ denote the test accuracy on the $t$-th task after the final incremental step. FAA is defined as:

$$\mathrm{FAA} = \frac{1}{T} \sum_{t=1}^{T} a^t$$

where $T$ is the total number of tasks. A higher FAA indicates better overall performance across all tasks and stronger continual learning ability.

# D   Results under Other Federated Incremental Learning Experimental Settings

In addition to the FCL setting proposed by Powder (ICML 2024)[19], which considers task overlaps over time, we evaluate our approach under the federated class incremental learning (FCIL) setting, as investigated by PILoRA (ECCV 2024)[80] and LoRM (ICLR 2025) [81].

**FCIL Setting:** The FCIL setting divides the learning process into 10 incremental tasks, where class distributions are disjoint across tasks. For each task, training data is distributed among 10 clients following a Dirichlet distribution with parameter $\beta \in \{0.5, 0.1, 0.05\}$ to simulate non-IID scenarios. A smaller $\beta$ value represents a stronger data imbalance among clients.

**Training Details:** A ViT-B/16 backbone pre-trained on ImageNet-21K is adopted. Each communication round consists of 5 training epochs, with a total of 5 communication rounds. Data augmentation for the training set includes random horizontal flipping and normalization. For the test set, preprocessing involves resizing with bicubic interpolation to $256 \times 256$, followed by center cropping to $224 \times 224$ and normalization.

**Comparison Results:** We compare our C$^2$Prompt with state-of-the-art FCIL methods [86; 87; 62; 11; 88; 89; 90; 91; 68; 80; 81] and the FCL method Powder [19] on the ImageNet-R benchmark. The Final Average Accuracy (FAA) [81] results are presented in Table 3. The results demonstrate that our C$^2$Prompt surpasses the state-of-the-art FCIL method LoRM, achieving improvements of **2.75%/5.91%/2.29%** at $\beta = 0.5/0.1/0.05$, respectively. Furthermore, compared to the state-of-the-art prompt-based FCL method Powder, our approach achieves **0.61%/2.60%/6.48%** improvements at $\beta = 0.5/0.1/0.05$, respectively. These increasing advantages under lower $\beta$ values are attributed to the effective inter-client intra-class distribution knowledge compensation mechanism, which significantly enhances model acquisition capacity and mitigates inter-client knowledge conflicts. These findings, alongside the experiments reported in the main paper, validate the adaptability of our approach to diverse practical federated continual learning scenarios.

# E   Experimental comparison on Cifar100

**Avg Comparison:** Only our C$^2$Prompt outperforms the state-of-the-art Powder, achieving improvements of **1.54%** on Cifar100. This finding highlights the advantage of our approach in long-term knowledge consolidation. This can be attributed to the substantial enhancement of new knowledge acquisition capability achieved through our local class distribution compensation and class-aware discriminativity prompt aggregation strategies. Additionally, the precise knowledge communication mechanism prevents the fusion of irrelevant prompts, which would otherwise produce invalid prompts that are not only semantically divergent from new prompts but also clash with historical prompts.

**AIA Comparison:** Our C$^2$Prompt achieves improvements of **0.86%** on Cifar100, confirming that our approach consistently outperforms existing methods across various training stages. These improvements are due to the local class distribution compensation and class-aware discriminativity

Table 3: Performance comparison on ImageNet-R with different $\beta$ values.

| Method | Publication | ImageNet-R (FAA) | | |
|---|---|---|---|---|
| | | $\beta = 0.5$ | $\beta = 0.1$ | $\beta = 0.05$ |
| EWC [86] | *NAS 2017* | 58.93 | 48.15 | 43.68 |
| LwF [87] | *PAMI 2017* | 54.03 | 41.02 | 46.07 |
| FisherAVG [88] | *NeurIPS 2022* | 58.68 | 50.82 | 47.33 |
| RegMean [89] | *ICLR 2023* | 61.18 | 57.00 | 55.80 |
| CCVR [90] | *NeurIPS 2021* | 70.00 | 62.60 | 60.38 |
| L2P [62] | *CVPR 2022* | 42.08 | 23.85 | 16.98 |
| CODA-P [11] | *CVPR 2023* | 61.18 | 36.73 | 25.82 |
| FedProto [91] | *AAAI 2022* | 58.52 | 47.30 | 52.93 |
| TARGET [68] | *ICCV 2023* | 54.65 | 45.83 | 41.32 |
| PILoRA [80] | *ECCV 2024* | 53.67 | 51.62 | 49.37 |
| Powder [19] | *ICML 2024* | 74.62 | 67.14 | 62.26 |
| LoRM [81] | *ICLR 2025* | 72.48 | 63.83 | 66.45 |
| C$^2$Prompt | *This Paper* | **75.23** | **69.74** | **68.74** |

Table 4: Performance comparison on CIFAR-100 with different $\beta$ values.

| Method | Publication | CIFAR-100 (FAA) | | |
|---|---|---|---|---|
| | | $\beta = 0.5$ | $\beta = 0.1$ | $\beta = 0.05$ |
| EWC [86] | *NAS 2017* | 78.46 | 72.42 | 64.51 |
| LwF [87] | *PAMI 2017* | 62.87 | 55.56 | 47.09 |
| FisherAVG [88] | *NeurIPS 2022* | 76.10 | 74.43 | 65.31 |
| RegMean [89] | *ICLR 2023* | 59.80 | 45.88 | 39.08 |
| CCVR [90] | *NeurIPS 2021* | 79.95 | 75.14 | 65.30 |
| L2P [62] | *CVPR 2022* | 83.88 | 61.54 | 55.00 |
| CODA-P [11] | *CVPR 2023* | 82.25 | 61.82 | 46.74 |
| FedProto [91] | *AAAI 2022* | 75.79 | 70.02 | 60.55 |
| TARGET [68] | *ICCV 2023* | 74.72 | 72.32 | 62.60 |
| PILoRA [80] | *ECCV 2024* | 76.48 | 75.81 | 74.80 |
| Powder [19] | *ICML 2024* | 87.46 | 85.33 | 82.03 |
| LoRM [81] | *ICLR 2025* | 86.95 | 81.76 | 82.76 |
| C$^2$Prompt | *This Paper* | **89.93** | **87.67** | **83.25** |

prompt aggregation designs, which strengthen robust local knowledge acquisition and enhance distributed knowledge collection.

**FM Comparison:** Our C$^2$Prompt shows a negative forgetting rate on the small-scale dataset Cifar100. This suggests that new tasks can facilitate the learning of historical tasks when training samples are limited. These results confirm the effective antiforgetting capability of our method.

**FT Comparison:** Our C$^2$Prompt shows advanced forward-transfer capacity, outperforming existing methods on Cifar100, respectively. This can be primarily attributed to two key components: the Global Class Distribution Estimation and Local Class Distribution Compensation mechanisms. Specifically, the former effectively leverages asynchronously arriving data from the same class to generate reliable global distribution estimates, while the latter utilizes these estimated global distributions to implement data-level information compensation, thereby significantly enhancing the learning efficiency of subsequent data.

**BT Comparison:** Our C$^2$Prompt consistently demonstrates positive backward transfer capability on the Cifar100 dataset. This arises from the fact that asynchronously arriving data allow subsequent tasks to enhance knowledge of previously seen classes. We observe that C²Prompt's backward-transfer results relatively outperform those of Fed-CODAP and Powder. This is because the Local Class Distribution Compensation and Class-aware Prompt Aggregation designs significantly boost

Table 5: Result comparison on the CIFAR-100 benchmark

| Methods | Publication | Avg↑ | AIA↑ | FM↓ | FT↑ | BT↑ | CT↑ |
|---|---|---|---|---|---|---|---|
| FedWEIT | ICML 2021 | 95.17 | 95.61 | 0.48 | 3.76 | -0.91 | 3.04 |
| CFeD | IJCAI 2022 | 73.87 | 79.06 | 2.07 | -11.01 | -4.71 | -14.78 |
| GLFC | CVPR 2022 | 95.35 | 95.92 | 0.35 | **5.51** | -0.54 | **5.08** |
| Fedspace | CVPR 2023 | 94.17 | 94.87 | 1.03 | 0.37 | -2.46 | -1.60 |
| Fed-L2P | CVPR 2022 | 95.65 | 95.68 | 0.08 | 0.89 | 0.08 | 0.95 |
| Fed-Dual | ECCV 2022 | 95.35 | 95.08 | 0.27 | 0.70 | -0.24 | 0.51 |
| Fed-CODAP | CVPR 2023 | 82.05 | 55.71 | 13.77 | -30.25 | -18.60 | -45.13 |
| FedCPrompt | ICML 2023 | 94.22 | 94.04 | 0.08 | 0.32 | 0.00 | 0.32 |
| Powder | ICML 2024 | 95.78 | 95.83 | 0.35 | 2.03 | -0.36 | 1.74 |
| PILoRA | ECCV 2024 | 76.21 | 82.31 | 0.32 | 0.01 | -0.42 | -0.40 |
| Fed-MOS | AAAI 2025 | 85.11 | 87.23 | 0.20 | -0.31 | -0.11 | -0.45 |
| LoRM | ICLR 2025 | 77.42 | 80.11 | 0.74 | 0.07 | 0.20 | 0.22 |
| Ours | This Paper | **97.32** | **96.78** | **-0.05** | 2.57 | **0.31** | 2.82 |

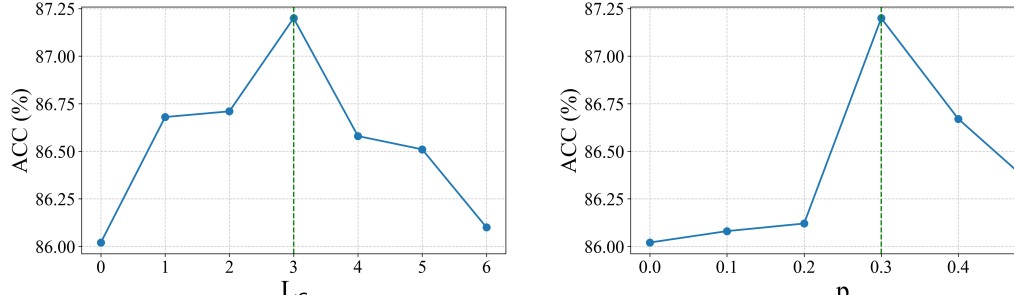

Figure 7: Ablation studies on the hyper-parameters under ImageNet-R dataset.

the distributed data learning capability at each stage, thereby leaving less improvement space for seen tasks.

**CT Comparison:** In terms of the comprehensive performance of forward and backward transfer, our C$^2$Prompt overall outperforms other existing methods that employ efficient fine-tuning. These results demonstrate that the class-aware client knowledge interaction designs proposed in this paper effectively enhance the overall learning capability of Federated Continual Learning (FCL) in the temporal dimension. Specifically: the Global Class Distribution Estimation module efficiently aggregates distributional information across spatial and temporal data sources; the Local Class Distribution Compensation module utilizes global distribution representations to overcome the non-IID (non-independent and identically distributed) phenomenon across clients; and the Local Discriminativity Learning and Class-aware Prompt Aggregation modules effectively integrate distributional knowledge into prompts.

# F    Analysis on the hyper-parameters

In Figure 7, we evaluate the performance of C$^2$Prompt under different values of the hyper-parameters $L_c$ and $p$. The parameter $L_c$ represents the length of the local class distribution compensation prompts. When the prompt length is less than or equal to 3, a larger value of parameter a enables the trained prompts to better fit the central distribution of the class, thereby improving the model's performance. Meanwhile, $p$ serves as the usage probability of local class distribution compensation prompts, is used to determine the number of generated new central distribution samples. Based on experimental analysis, the optimal hyperparameter value for $p$ is set to 0.3.

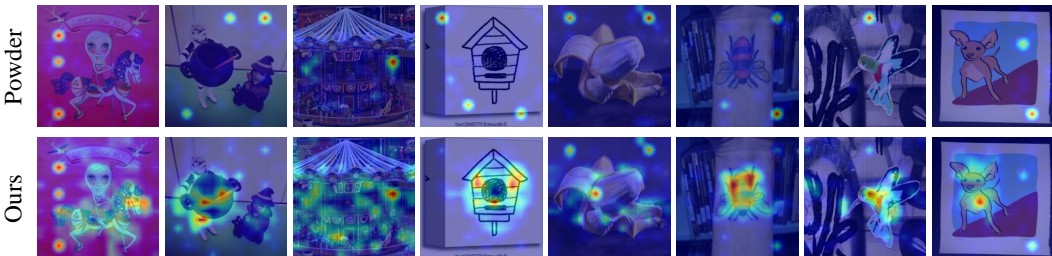

Figure 8: Prompt attention visualization on ImageNet-R.

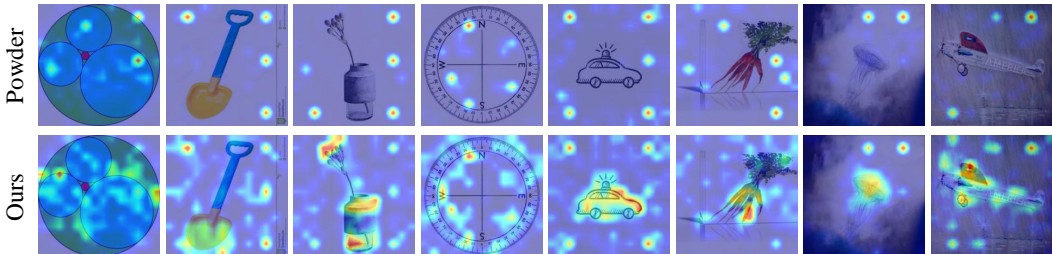

Figure 9: Prompt attention visualization on DomainNet.

## G Prompt attention visualization comparison on different benchmarks

Figure 8 and Figure 9 present the visualization comparison of prompt attention maps between our $C^2$Prompt framework and the state-of-the-art Powder method across the challenging ImageNet-R and DomainNet benchmarks. The results demonstrate that prompts generated by Powder are predominantly influenced by class-irrelevant knowledge, leading to limited discriminative feature extraction capabilities. In contrast, our method effectively focuses prompt activations on discriminative regions while significantly reducing interference from class-agnostic knowledge. These improvements are primarily attributed to the proposed Class-aware Prompt Aggregation mechanism, which systematically alleviates the fusion of knowledge conflicts during prompt aggregation through explicit semantic alignment.

## H Broader Impacts

Our method tackles a practical federated continual learning (FCL) problem and introduces a novel approach that effectively improves the local parameter learning in the client side and enhances knowledge aggregation capacity on the server side.

**The Potential Positive Societal Impacts of this research include:**

*1. Enhanced Privacy Preservation in Decentralized Learning*

Federated continual learning (FCL) inherently avoids centralized data collection. $C^2$Prompt further eliminates reliance on raw data or generative models for knowledge retention, reducing risks of sensitive data leakage. This is critical for applications like healthcare (e.g., personalized disease prediction across hospitals) or finance (e.g., fraud detection without sharing transaction details).

*2. Improved Adaptability in Dynamic Environments*

By addressing both temporal and spatial forgetting, $C^2$Prompt enables models to continuously adapt to evolving data streams. This ensures long-term reliability in scenarios where data distributions shift over time or vary across regions.

*3. Democratization of AI in Resource-Constrained Settings*

The lightweight prompt-based framework reduces computational and communication overhead compared to traditional methods. This democratizes access to AI for edge devices with limited

resources (e.g., rural IoT sensors, low-power medical devices), fostering equitable technological progress.

### 4. Mitigation of Model Bias via Class-Aware Aggregation

The class-aware prompt aggregation (CPA) mechanism explicitly accounts for inter-client class relevance, potentially reducing biases arising from skewed local data distributions. For instance, in facial recognition systems deployed across diverse demographics, CPA could improve fairness by ensuring minority groups' features are adequately represented.

**The Potential Negative Societal Impacts of this research include:**

### 1. Energy Consumption

Additional distributional information communication and client-wise aggregation across distributed clients may increase energy consumption, particularly in large-scale deployments.

