# OpenReview forum: "C$^2$Prompt: Class-aware Client Knowledge Interaction for Federated Continual Learning"
_NeurIPS.cc/2025/Conference — NeurIPS 2025 poster_

### Official Review · Reviewer_Zi11 · 2025-06-06

**Clarity:** 3
**Significance:** 3
**Originality:** 3
**Rating:** 4
**Confidence:** 4

**Summary:**

This paper proposes C²Prompt, a method for federated continual learning (FCL) that enhances class-wise knowledge coherence across distributed clients to mitigate both temporal and spatial forgetting. The key insight is that prompt-based FCL suffers from intra-class distribution gaps and inter-class knowledge confusion during prompt communication. To address this, the authors introduce two core components: a local class distribution compensation (LCDC) mechanism to align intra-class semantics across clients, and a class-aware prompt aggregation (CPA) strategy to preserve discriminative, class-relevant knowledge. By explicitly modeling and resolving class-level inconsistencies, C²Prompt enables more robust and coherent learning across clients. Experimental results on standard FCL benchmarks show that the proposed method achieves state-of-the-art performance.

**Questions:**

1. Can the authors provide additional qualitative or quantitative evidence to directly support the claim that C²Prompt improves both intra-class distribution alignment and inter-prompt class-wise relevance?

2. What is the specific contribution of the knowledge distillation (KD) loss (from Powder [19]) to the overall performance? Could you provide an ablation that excludes KD to isolate the gains from LCDC and CPA?

3. Why is the local class distribution compensation (LCDC) prompt retrieved using the label y, rather than being generated through the instance-specific weighted combination strategy used for the local discriminativity prompts?

**Ethical Concerns:**

["NO or VERY MINOR ethics concerns only"]

**Final Justification:**

Thanks for the detailed rebuttal, which has addressed most of my concerns. Given these clarifications and the overall strength of the proposed method, I will maintain my recommendation, now with increased confidence in the paper’s contributions.

**Limitations:**

Yes.

**Paper Formatting Concerns:**

No.

**Quality:**

3

**Strengths And Weaknesses:**

Strengths
1. The paper presents a well-motivated and thoughtfully designed prompt-based approach for FCL, with a clear emphasis on improving class-wise knowledge coherence. The proposed components (LCDC and CPA) are effectively integrated, leading to strong empirical performance that surpasses recent competitive baselines.

2. The paper is well-written and easy to follow.

Weaknesses
1. While the method is shown to outperform prior approaches, the claim that C²Prompt improves both inter-prompt class-wise relevance and intra-class distribution alignment would be stronger with more direct qualitative or quantitative evidence. For example, visualizations (e.g., t-SNE plots) or class-wise similarity metrics could help demonstrate the improvements in class coherence more explicitly.

2. In addition to the proposed LCDC and CPA components, the method incorporates the knowledge distillation (KD) loss introduced by Powder [19]. However, the paper lacks an ablation study isolating the impact of the KD loss. As a result, it remains unclear how much of the performance gain over baselines (especially beyond Powder) stems from the new components versus the KD loss. A more detailed ablation would significantly strengthen the empirical validation.

3. The mechanism by which the local class distribution compensation prompt is retrieved from the pool using the label y warrants further justification. Specifically, why is it not generated via the same instance-specific weighted-sum strategy used for the local discriminativity prompts? Clarifying this design choice could help readers better understand the rationale and benefits of using explicit label-based indexing in this context.

4. Typo: “the ocal class distribution” -> “the local class distribution”.

---

> ### Author Rebuttal · Authors · 2025-07-31
>
> Thank you for your constructive feedback and recognition. Below are our responses, which we hope effectively address your concerns.
>
>
> **W1: Improvements in class coherence**
>
> We appreciate your insightful suggestion. As visualizations are not supported during the rebuttal phase, we instead provide quantitative evidence to demonstrate how our method enhances inter-prompt class-wise relevance and intra-class distribution alignment.
>
> (1) Mean Inter-Prompt Class-wise Relevance Score (mPR).
>
> The mPR measures the overall class-wise relevance among prompts during fusion. It is defined as:
> $$mPR=1/N_p\sum_{i=1}^{N_p}(\sum_{j=1}^C\sum_{k=1}^{N_p}s_{i,j}W_{i,k}s_{k,j})$$
> where $N_p$ and $C$ are the prompt number and class number in a task, respectively. $s_{i,j}$ is the proportion of class $j$'s matched instance for prompt $i$. $W_{i,k}$ denotes the weight of prompt $k$ when generating the fusion prompt corresponding to prompt $i$.
>
> The mPR comparison of our method and the state-of-the-art Powder is shown as follows:
>
> |Method|Powder|Ours|
> |-|-|-|
> |mPR↑|0.124|0.160|
>
> Our method achieves a **0.036** absolute gain in mPR, corresponding to a **29.0%** relative improvement over Powder. This result verifies that our Class-aware Prompt Aggregation design effectively filters and fuses class-relevant prompts, leading to better semantic alignment. As a result, our method achieves superior new knowledge acquisition and anti-forgetting capacity, as shown in Figure 1(a) of the main paper.
>
> (2) Mean Intra-Class Distribution Distance (mCD).
>
> mCD assesses the distributional alignment between local and global class representations. It is calculated as:
>
> $$mCD=\frac{1}{KC}\sum_{k=1}^{K}\sum_{j=1}^CKL(d_j, d_{k,j})$$
> where $K$ is the client number, $ d_i$ is the ground truth distribution of class $j$, which is a Gaussian distribution obtained by calculating the mean and variance of the instance features of class $j$. $d_{k,j}$ is the local distribution of class $j$ on client $k$. $d_{k,j}$ is also a Gaussian distribution. For existing methods, $d_{k,j}$ is obtained by calculating the mean and variance of class $j$ using data of client $k$. In our method, $d_{k,j}$ is a fused distribution of the class-wise local data and transferred data.
>
> The quantitative comparison of our method and Powder under the mCD metric is reported as follows:
>
> |Method|Powder|Ours|
> |-|-|-|
> |mCD↓|7.67|6.98|
>
> Our method reduces mCD by **0.69** compared to Powder, which is a **9%** improvement, demonstrating that the proposed Local Class Distribution Compensation module effectively narrows the gap between local and global distributions. This promotes inter-client knowledge consistency and enhances the model's overall knowledge accumulation, as shown in Figure 1(b) of our main paper.
>
>
> **W2: Ablation on the KD loss**
>
> (1) The KD loss is an inherent component of the Powder baseline. Therefore, the performance improvement reported in our experiments is solely attributed to our proposed LCDC and CPA modules. Specifically, as illustrated in Figure 4 of our main paper, LCDC and CPA individually contribute **1.88%** and **1.33%** improvements, respectively. When combined, a total gain of **2.51%** is observed.
>
> (2) To further validate the effectiveness of our proposed modules, we conduct experiments by removing the KD loss used in Powder. The results are presented below:
>
>  |Method|Powder-w/o KD|Ours-w/o KD|
> |-|-|-|
> |Avg|77.63|79.22|
> |AIA|76.01|76.62|
>
> Even without KD, our method consistently outperforms the Powder baseline, achieving **1.59%** on Avg and **0.61%** on AIA. These results further confirm the robustness of our design, independent of KD.
>
> (3) In fact, KD is a **widely adopted fundamental technique** in federated continual learning methods, such as Podnet [a], CFeD [b], GLFC [c], FedSpace [d], and Powder [e]. It is typically employed to align the aggregated global model with local client models, especially under non-IID task distributions. Without KD, local models tend to overfit to new data and forget previously learned global knowledge. Therefore, the use of KD is orthogonal and complementary to our contributions.
>
> [a] Podnet: Pooled Outputs Distillation for Small-Tasks Incremental Learning. ECCV 2020
>
> [b] Continual Federated Learning based on Knowledge Distillation. IJCAI 2022
>
> [c] Federated Class-Incremental Learning. CVPR 2022
>
> [d] Asynchronous Federated Continual Learning. CVPR 2023
>
> [e] Federated Continual Learning via Prompt-based Dual Knowledge Transfer. ICML 2024
>
>
> **W3: The design choice of local class distribution compensation prompt**
>
> (1) We retrieve the local class distribution compensation prompt from the pool using the label $y$ to **avoid inter-class interference** during prompt training.
>
> (i) Specifically, local distributions for each class typically exhibit **independent and random shifts** from the global distribution. As a result, there are no unified semantic patterns across classes for guiding distribution-shift modeling. Using a shared prompt pool for all classes, followed by weighted fusion, introduces **optimization conflicts** across classes and leads to sub-optimal learning.
>
> (ii) In contrast, our method explicitly utilizes the class label $y$ to segregate prompts for different classes, enabling **focused and optimal learning** of class-specific prompts.
>
>
> (2) We further compare our method with a weighted-sum prompt strategy. The results are as follows:
>
>  |Method|weighted-sum|Ours|
> |-|-|-|
> |Avg|86.30|87.20|
> |AIA|85.72|85.93|
>
> Our approach surpasses the weighted-sum baseline by **0.90%** on Avg and **0.21%** on AIA, confirming the effectiveness of our class-specific compensation prompt design in capturing localized distribution shifts in federated scenarios.
>
>
> **W4: Typo**
>
> Thank you for identifying this issue. We have corrected "the ocal class distribution" to "the local class distribution" in the revised version.
>
> We sincerely appreciate your thoughtful and constructive feedback, which has significantly contributed to strengthening our work. We hope our response sufficiently addresses the concerns raised, and we would be grateful if the reviewer considers increasing the rating. Please feel free to raise any additional questions.

---

> > ### Comment · Reviewer_Zi11 · 2025-08-04
> >
> > Thanks for the detailed rebuttal, which has addressed most of my concerns. Given these clarifications and the overall strength of the proposed method, I will maintain my recommendation, now with increased confidence in the paper’s contributions.

---

> > > ### Author Response · Authors · 2025-08-04
> > >
> > > Dear Reviewer Zi11,
> > >
> > > Thank you sincerely for your positive evaluation and increased confidence in our contributions. Your insightful feedback has been invaluable in highlighting the novelty and effectiveness of our work. We will diligently incorporate the suggested additional experiments and discussion into the final version to enhance its comprehensiveness.
> > >
> > > Best regards,
> > >
> > > The Authors

---

### Official Review · Reviewer_nSnG · 2025-06-17

**Clarity:** 3
**Significance:** 3
**Originality:** 3
**Rating:** 5
**Confidence:** 5

**Summary:**

This paper addresses the Federated continual learning (FCL) task, where the key challenges lie in temporal forgetting over time and spatial forgetting. The authors underscore that state-of-the-art prompt-based FCL methods are prone to class-wise knowledge coherence between prompts across clients. The class-wise knowledge coherence includes intra-class distribution gap across clients and inter-prompt class-wise relevance. To address these issues, a novel method, Class-aware Client Knowledge Interaction (C$^2$Prompt), is proposed to enhance class-wise knowledge coherence during prompt communication. Extensive experiments on multiple FCL benchmarks demonstrate the effectiveness of the proposed method.

**Questions:**

The robustness and generalizability of the method could be better demonstrated by:
1. Including results on additional benchmarks (e.g., ImageNet-A and CIFAR-100) under federated class-incremental learning (FCIL) settings.
2. Providing a clearer definition and justification of the Final Average Accuracy (FAA) metric.

**Ethical Concerns:**

["NO or VERY MINOR ethics concerns only"]

**Final Justification:**

Thank the authors for their thorough rebuttal. My previous concerns have been satisfactorily addressed by the additional experiments and detailed discussion provided. Furthermore, after reviewing the feedback from the other reviewers and the authors' responses, I believe this work is highly meaningful and provides valuable insights to the federated continual learning community. Accordingly, I am raising my score from 4 to 5.

**Limitations:**

Yes.

**Quality:**

3

**Strengths And Weaknesses:**

**Strength:**
- The structure of this paper is well-organized with clear writing and well-designed figures that effectively illustrate methodological innovations and superiority.
- This paper is well-motivated and introduces a novel method to address the FCL problem. The analysis of class-wise knowledge coherence issues is insightful in FCL. The class distribution-guided local distribution compensation and server-side prompt-aggregation designs are innovative and reasonable.
- The design of LCDC and CPA is well-justified with clear mathematical discussion and theoretical support in Appendix A. The distribution aggregation and compensation mechanism is especially thoughtful.
- The proposed method achieves state-of-the-art results on multiple FCL benchmarks, particularly in average accuracy and AIA metrics. Besides, the ablation studies and performance curves are informative and verify the effectiveness of each component. Communication and parameter overheads are also thoroughly analyzed.

**Weakness:**
- Limited FCIL Benchmark Diversity: Under the federated class-incremental learning setting, only ImageNet-R results are provided. Expanding comparisons to ImageNet-A and CIFAR-100 would better demonstrate the generalizability of the proposed method.
- Undefined Metric: Final Average Accuracy (FAA) metric in the FCIL setting requires explicit definition and differentiation from related metrics (Avg, AIA, FM, FT, BT, CT).

---

> ### Author Rebuttal · Authors · 2025-07-31
>
> We sincerely appreciate the reviewer's constructive feedback and recognition. Please find our point-by-point responses below.
>
> **W1: More experimental comparison on FCIL benchmarks**
>
> Thank you for the valuable suggestion. We have conducted additional experiments on CIFAR-100 and ImageNet-A under the Federated Continual Incremental Learning (FCIL) setting [a], using the Dirichlet distribution with parameter $\beta$ to control data heterogeneity. The results are presented below:
>
> |CIFAR-100|$\beta$=0.5|$\beta$=0.1|$\beta$=0.05|
> |-|-|-|-|
> |LoRM|86.95|81.76|82.03
> |Powder|87.46|85.33|82.76
> |Ours|89.93|87.67|83.25
>
> Our method outperforms the state-of-the-art FCIL method LoRM by **2.98%/5.91%/1.22%** at $\beta=0.5 / 0.1 / 0.05$, respectively. Compared to the prompt-based FCL baseline Powder, we achieve **2.47%/2.34%/0.49%** improvements under the same settings.
>
>
> |ImageNet-A|$\beta$=0.5|$\beta$=0.1|$\beta$=0.05|
> |-|-|-|-|
> |LoRM|37.26|36.34|33.11
> |Powder|39.29|37.11|35.62
> |Ours|40.51|39.76|37.21
>
> Similarly, our method surpasses LoRM by **3.25%/3.42%/4.10%**, and Powder by **1.22%/2.65%/1.59%** at $\beta=0.5 / 0.1 / 0.05$, respectively.
>
> In summary, these results validate the robustness and adaptability of our method under varying levels of data heterogeneity. In particular, the growing performance gains under smaller $\beta$ values can be attributed to our inter-client, intra-class distribution compensation mechanism, which enhances the model's ability to acquire and aggregate discriminative knowledge while mitigating inter-client conflicts.
>
> [a] Closed-form merging of parameter-efficient modules for federated continual learning. ICLR 2025
>
> **W2: Definition of FAA**
>
> Thanks for pointing this out.
>
> (1) Final Average Accuracy (FAA) is a standard metric used in FCIL to measure knowledge retention and accumulation. Let $a^t$ denote the test accuracy on the $t$-th task after the final incremental step. FAA is defined as:
> $$\mathrm{FAA}=\frac{1}{T}\sum_{t=1}^Ta^t$$
> where $T$ is the total number of tasks. A higher FAA indicates better overall performance across all tasks and stronger continual learning ability.
>
> (2) We have incorporated the definition of FAA into our appendix to improve the clarity and comprehensiveness of our paper.
>
> We hope our responses adequately address your concerns. Please feel free to reach out with any further questions or suggestions.

---

> > ### Comment · Reviewer_nSnG · 2025-08-01
> >
> > Thank the authors for their thorough rebuttal. My previous concerns have been satisfactorily addressed by the additional experiments and detailed discussion provided. Furthermore, after reviewing the feedback from the other reviewers and the authors' responses, I believe this work is highly meaningful and provides valuable insights to the federated continual learning community. Accordingly, I am raising my score from 4 to 5.

---

> > > ### Author Response · Authors · 2025-08-04
> > >
> > > Dear Reviewer nSnG,
> > >
> > > We are pleased that our responses have addressed your concerns and sincerely appreciate your increased rating. Your constructive comments have significantly strengthened our manuscript.
> > >
> > > Best regards,
> > >
> > > The Authors

---

### Official Review · Reviewer_gSYz · 2025-06-26

**Clarity:** 3
**Significance:** 3
**Originality:** 3
**Rating:** 4
**Confidence:** 3

**Summary:**

This paper addresses the challenges of spatio-temporal forgetting in Federated Continual Learning (FCL) by enhancing class-wise knowledge coherence across clients. The authors propose C2Prompt, a prompt-based framework introducing two novel modules: (1) Local Class Distribution Compensation to mitigate intra-class distribution disparity, and (2) Class-aware Prompt Aggregation to reduce inter-class knowledge conflict. Experiments demonstrate the effectiveness of the proposed method.

**Questions:**

The authors are expected to address the concerns in the "weaknesses".

**Ethical Concerns:**

["NO or VERY MINOR ethics concerns only"]

**Limitations:**

yes

**Quality:**

3

**Strengths And Weaknesses:**

Strengths:
1. The paper identifies two underexplored but crucial sources of knowledge degradation in prompt-based FCL: intra-class distribution gaps and inter-prompt class relevance.
2. The proposed method outperforms several state-of-the-art approaches across six evaluation metrics.
3. The paper is well written and easy to follow.
4. Theoretical justification is provided to further demonstrate the usefulness of the designs in the proposed paper.

Weaknesses:
1. More datasets are expected for a comprehensive comparison.
2. The efficiency and communication costs of the proposed method (compared to baselines) can be further discussed.
3. The experiments on scenarios with severe data heterogeneity are expected.

---

> ### Author Rebuttal · Authors · 2025-07-31
>
> Thank you for your valuable feedback and insightful comments. We address your concerns as follows:
>
> **W1: More datasets for comparison**
>
> (1) In the main paper, we report experimental results on DomainNet and ImageNet-R in Table 1, following the settings in Powder [a].
>
> (2) In **Table 4** of our appendix, we evaluate our method on the **CIFAR-100** dataset. Our approach outperforms the state-of-the-art method Powder by **1.54%** on the Avg metric and **0.86%** on the AIA metric.
>
> (3) We further conduct experiments on the ImageNet-A dataset. The results are summarized below:
> |ImageNet-A|Avg|AIA|
> |-|-|-|
> |Powder|66.35|63.14
> |Ours|68.34|65.34
>
> Our method surpasses Powder by **1.99%** on Avg and **2.20%** on AIA, further confirming its robustness on challenging real-world benchmarks.
>
> (4) These consistent improvements across multiple benchmarks verify the adaptability and generalizability of our approach. The performance gain primarily stems from our two core designs:
>
> (i) Local Class Distribution Compensation (LCDC) actively models the global distribution of each class, thereby **improving local discriminative knowledge acquisition and enhancing inter-client knowledge compatibility**. This results in a more robust and informative prompt aggregation process.
>
> (ii) Class-aware Prompt Aggregation (CPA) selectively filters and preserves class-specific prompts, effectively **mitigating inter-class knowledge interference during aggregation**. This design effectively consolidates the discriminative capacity of the learned features across clients.
>
> Together, these modules enable effective distribution modeling at both global and local levels, enabling our model to boost performance on diverse benchmarks.
>
> [a] Federated Continual Learning via Prompt-based Dual Knowledge Transfer. ICML 2024
>
> **W2: Efficiency and communication costs**
>
> We report communication and parameter overhead in **Table 2** of the main paper. The results show our method maintains strong efficiency:
>
> (1) Communication Overhead:
>
> Our method requires 496.01MB for communication. Compared with prior prompt-based methods (FED-L2P, FED-DUAL, Fed-CODAP, Fed-CPrompt), this is **125.77MB to 319.62MB** lower, yielding a **25.4% to 64.4% reduction**. Compared to Powder, our method incurs only **2.93MB (0.6%)** additional overhead, mainly due to the exchange of lightweight class distribution information. Since the number of classes is small, this additional cost is negligible.
>
> (2) Training Overhead:
>
> Our method introduces 2.82MB of learnable parameters. Compared with other baselines, this is **1.14MB to 8.61MB fewer**, leading to **40.4% to 305.3%** better training efficiency. Compared with Powder, our method adds only **0.18MB (6.8%)**, resulting from the lightweight distribution prompts introduced by the LCDC module.
>
> (3) Inference Overhead:
>
> At inference time, our method requires only **2.64MB** of additional parameters, which is the same as Powder and **50% to 333.0%** more efficient than other prompt learning baselines. This is because only discriminative prompts are retained during inference, incurring **no additional cost compared to Powder**.
>
> **W3: Experiments with severe data heterogeneity**
>
> (1) In Table 3 of our appendix, we report experimental results on the ImageNet-R dataset under varying levels of data heterogeneity using the Dirichlet distribution parameter $\beta$ ranging from 0.5 to 0.05. Notably, $\beta=0.05$ is typically regarded as a setting with severe data heterogeneity [b]. Under this condition, our method achieves a **2.29%** performance gain over the state-of-the-art LoRM [b], and a substantial **6.48%** improvement compared to the prompt-based baseline Powder.
>
> [b] Closed-form merging of parameter-efficient modules for federated continual learning. ICLR 2025
>
> (2) We further evaluate our method on the CIFAR-100 and ImageNet-A datasets under various levels of data heterogeneity. The results are summarized below:
>
> |CIFAR-100|$\beta$=0.5|$\beta$=0.1|$\beta$=0.05|
> |-|-|-|-|
> |LoRM|86.95|81.76|82.03
> |Powder|87.46|85.33|82.76
> |Ours|89.93|87.67|83.25
>
> The results demonstrate that our method surpasses the state-of-the-art FCIL method LoRM, achieving improvements of **2.98%/5.91%/1.22%** at $\beta = 0.5/0.1/0.05$, respectively. Furthermore, compared to the state-of-the-art prompt-based FCL method Powder, our approach achieves **2.47%/2.34%/0.49%** improvements at $\beta = 0.5/0.1/0.05$, respectively.
>
>
> |ImageNet-A|$\beta$=0.5|$\beta$=0.1|$\beta$=0.05|
> |-|-|-|-|
> |LoRM|37.26|36.34|33.11
> |Powder|39.29|37.11|35.62
> |Ours|40.51|39.76|37.21
>
> On ImageNet-A, our method surpasses LoRM by **3.25%/3.42%/4.10%** and Powder by **1.22%/2.65%/1.59%** at $\beta = 0.5/0.1/0.05$, respectively.
>
> (3) These results clearly demonstrate that our method maintains robust performance across varying degrees of data heterogeneity, particularly under severe heterogeneity conditions (i.e., small $\beta$ values). The advantages over the Powder baseline in lower $\beta$ regimes can be attributed to our inter-client, intra-class distribution compensation mechanism, which effectively aligns client-specific distributions, enhances the model's knowledge acquisition capability, and reduces inter-client knowledge conflicts.
>
>
> We sincerely appreciate the reviewer's thoughtful and constructive feedback, which has significantly contributed to strengthening our work. We hope our response sufficiently addresses the concerns raised, and we would be grateful if the reviewer considers increasing the rating. Please feel free to raise any additional questions.

---

> > ### Comment · Reviewer_gSYz · 2025-08-07
> > **Response to Authors**
> >
> > Thanks for your additional experiments on multiple datasets, efficiency, and severe data heterogeneity. It addresses my corresponding concerns. I would like to maintain my score.

---

### Official Review · Reviewer_nsjy · 2025-06-28

**Clarity:** 2
**Significance:** 3
**Originality:** 2
**Rating:** 4
**Confidence:** 5

**Summary:**

This paper proposes C$^2$Prompt, a novel class-aware client knowledge interaction method for FCL. It addresses the key challenges of temporal forgetting and spatial forgetting by enhancing class-wise knowledge coherence across distributed clients. The approach involves two main components: a local class distribution compensation mechanism (LCDC), which reduces intra-class distribution disparities among clients to improve intra-class semantic consistency; and a class-aware prompt aggregation scheme (CPA), which estimates class-wise relevance to better align prompts globally and mitigate knowledge conflicts. Extensive experiments on multiple FCL benchmarks demonstrate that C$^2$Prompt significantly outperforms existing methods.

**Questions:**

1.I want to know whether the uploaded local class-aware feature distributions will cause privacy leakage.

2.Why are the locations of indiscriminative attention of the proposed method similar to those of the Powder in Figure 5? For example, the two points above the first image.

**Ethical Concerns:**

["NO or VERY MINOR ethics concerns only"]

**Final Justification:**

Thanks for the clarification and additional evaluation. The authors' rebuttal has addressed my concerns, so I will increase my score to 4.  I recommend incorporating these experiments into the final version, as this would make the paper more comprehensive and persuasive.

**Limitations:**

Yes

**Paper Formatting Concerns:**

No major formatting issues.

**Quality:**

3

**Strengths And Weaknesses:**

Strengths:

1.The paper thoroughly discusses the issues of knowledge coherence in federated prompts and provides well-motivated mechanisms to address these, supported by theoretical considerations.

2.The paper considers an important scenario: continuous learning in the federated scenario.

Weaknesses:

1.The layout needs to be corrected. For example, Figure 5 appears before Figure 3. Figure 6 is mentioned before Figure 5 in Line 319.

2.There are many clerical errors. For example, Fig. 2 (a) and Fig. 2 (c) in Lines 303 and 305.

3.Lack of experiments on different models. This paper only considered one model, i.e., ViT-B/16 backbone pre-trained on ImageNet-21K. It is better to consider the impact of larger or smaller model architectures and out-of-distribution pre-training datasets.

4.Equation 13 remains rather unclear. What is the relationship between the prediction of an image and $L_{ce}$ during the inference stage?

5.In the experiments, the training tasks are sampled from each dataset.  It is better to investigate the scenario where different tasks are sampled across different datasets. For example, task 1 is a 20-classification task sampled from ImageNet-R, and task 2 is a 10-classification task sampled from CIFAR-100.

---

> ### Author Rebuttal · Authors · 2025-07-31
>
> Thanks for the valuable feedback. We hope our responses effectively address your concerns.
>
> **W1: Layout correction**
>
> Thank you for the suggestion. We have revised the manuscript layout to ensure that all figures appear in the correct order and align properly with the corresponding textual references.
>
> **W2:  Clerical errors**
>
> Thank you for pointing this out. The references to Fig. 2(a) and Fig. 2(c) in Lines 305–306 should have been Fig. 2(b) and Fig. 2(d), respectively. We have corrected this typo in the revised version.
>
> **W3-1: Larger or smaller model architectures**
>
> (1) Larger Architectures:
>
> We have adopted ViT-L as the backbone and compared our method with the state-of-the-art Powder. The results are as follows:
>
> |DomainNet/ViT-L|Avg|AIA|
> |-|-|-|
> |Powder|77.75|77.11|
> |Ours|79.55|77.72
>
> |ImageNet-R/ViT-L|Avg|AIA|
> |-|-|-|
> |Powder|87.52|85.79
> |Ours|87.94|85.80
>
> Our method achieves **1.80%**/**0.61%** improvements on DomainNet and **0.42%**/**0.01%** improvement on ImageNet-R under the Avg/AIA metrics compared to Powder.
>
> The more significant improvement on DomainNet can be attributed to its larger domain gap from the pretraining dataset (ImageNet-21K). In this case, our class-aware client knowledge interaction mechanism improves knowledge acquisition by **enhancing new domain distribution adaptation**.
>
> In contrast, the relatively modest gain on ImageNet-R is due to the strong alignment between ImageNet-R and the pretraining dataset. ViT-L can already capture a large portion of relevant knowledge during pretraining, leaving limited room for post-training improvement. Nevertheless, the consistent performance gains on both benchmarks verify the effectiveness of our method when using large-scale architectures.
>
> (2) Smaller Architectures:
>
> We have also evaluated our method using ViT-S as the backbone. The results are shown below:
>
> |DomainNet/ViT-S|Avg|AIA|
> |-|-|-|
> |Powder|75.16|73.80
> |Ours|76.44|75.47
>
> |ImageNet-R/ViT-S|Avg|AIA|
> |-|-|-|
> |Powder|81.92|81.47|
> |Ours|83.43|82.69
>
> Our method outperforms Powder by **1.28**/**1.67** on DomainNet and **1.51**/**1.22** on ImageNet-R under Avg/AIA metrics, demonstrating its adaptability to small-scale architectures.
>
> (3) Summary:
>
> These results confirm that our approach is **robust across different model scales** and consistently outperforms the state-of-the-art Powder. This is mainly because our class-aware client knowledge interaction mechanism leverages distributional information, which is largely independent of the backbone scale.
>
> **W3-2: Out-of-distribution pre-training datasets**
>
> (1) We would like to clarify that all reported benchmarks, DomainNet, ImageNet-R, and CIFAR-100, **are out-of-distribution datasets relative to the pre-training dataset ImageNet-21K** [a]. This is because part (ImageNet-R) or all (DomainNet, CIFAR-100) of their images do not appear in ImageNet-21K, and they typically exhibit significant domain shifts from the pre-training distribution.
>
> [a] What Variables Affect Out-of-Distribution Generalization in Pretrained Models? NeurIPS 2024
>
> (2) To further address the reviewer's concern, we additionally evaluate our method in another out-of-distribution setting by adopting **CLIP** as the backbone and conducting post-training on **unseen benchmarks** [b].
>
> [b] SCAP: Transductive Test-Time Adaptation via Supportive Clique-based Attribute Prompting. CVPR 2025
>
> The results are presented below:
> |DomainNet/CLIP|Avg|AIA|
> |-|-|-|
> |Powder|82.29|82.43|
> |Ours|84.18|83.71
>
> |ImageNet-R/CLIP|Avg|AIA|
> |-|-|-|
> |Powder|89.44|89.53
> |Ours|90.48|90.01
>
> Our method achieves **1.89%**/**1.28%** Avg/AIA improvements on DomainNet and **1.04%**/**0.48%** on ImageNet-R, respectively. These results demonstrate that our method is not dependent on the pre-training distribution and generalizes well under out-of-distribution scenarios.
>
> This robustness is primarily attributed to the fact that many high-level visual semantics (such as shape, structure, and layout) are transferable across domains. Our Local Distribution Compensation and Class-Weighted Prompt Aggregation modules are capable of ** exploiting such domain-invariant cues**.  By modeling the global distribution and adjusting local distributions on clients, they enable effective adaptation from pre-training models to unseen target domains in the federated continual learning scenarios.
>
> **W4: Equation 13**
>
> Thank you for pointing this out. The notation $\mathcal{L}_{ce}$ in Equation 13 is indeed a typo. It should be corrected to $\hat{y}$, which denotes the predicted class probabilities generated by the model during inference. We have carefully corrected this issue in the revised version.
>
> **W5: Cross-dataset sampling**
>
> (1) We have conducted additional experiments by alternately sampling task classes from ImageNet-R and CIFAR-100, as suggested. The results are summarized below:
>
> |ImageNet-R+CIFAR-100|Avg|AIA|
> |-|-|-|
> |Powder|90.80|92.68|
> |Ours|93.18|93.31|
>
> Our method achieves **93.18%** and **93.31%** on the Avg and AIA metrics, respectively, outperforming the state-of-the-art Powder by **2.38%** and **0.63%**. These results demonstrate that our approach generalizes effectively under the more challenging cross-dataset federated incremental learning scenario. This robustness partially stems from the prompt learning scheme, which helps disentangle knowledge across datasets and mitigates mutual interference. In addition, the performance gain over Powder is primarily attributed to our Local Class Distribution Compensation and Class-Weighted Prompt Aggregation designs, which enhance the model's ability to learn and accumulate knowledge from clients at each stage.
>
> **Q1: Privacy leakage**
>
> The uploaded local class-aware feature distributions **do not** cause privacy leakage
>
> (1) The uploaded distributions only contain the mean and variance of each local class. **No instance-level information or raw data** is transmitted to the server, ensuring that no individual-specific information is exposed.
>
> (2) These uploaded statistics are equivalent to distributional prototypes, a widely adopted strategy in **privacy-preserving learning**. Similar approaches have been employed in existing privacy-friendly methods such as [c,d].
>
> [c] Fecam: Exploiting the heterogeneity of class distributions in nonexemplar continual learning. NeurIPS 2024
>
> [d] Distribution-aware knowledge prototyping for non-exemplar lifelong person re-identification. CVPR 2024
>
> **Q2: Indiscriminative attention**
>
> (1) The presence of indiscriminative attention patterns similar to those in Powder is expected, as Powder serves as our baseline. Our method inherits the local discriminative prompt from Powder, which may include suboptimal attention behaviors. However, our approach mitigates the impact of indiscriminative knowledge by enhancing the acquisition and accumulation of discriminative information as discussed below.
>
> (2) We further observe that the locations of indiscriminative attention tend to be shared across different images, e.g., the two hotspots above the first, second, and sixth images. This phenomenon stems from dataset bias, where certain non-informative regions frequently appear in the same spatial positions across images, thus leading to spuriously high attention scores.
>
> (3) Our method significantly **reduces such indiscriminative attention** through the following designs:
>
> (i) The Local Class Distribution Compensation module actively models the global distribution of each class. This **enhances both intra-client discriminative knowledge learning and inter-client knowledge compatibility**, thereby facilitating more discriminative knowledge aggregation.
>
> (ii) The Class-aware Prompt Aggregation mechanism selectively retains class-specific prompts while avoiding inter-class knowledge conflicts during aggregation. This strengthens discriminative feature extraction and **suppresses noisy or irrelevant information**.
>
> In summary, our proposed method primarily overcomes the influence of indiscriminative attention inherited from the baseline by significantly improving the model's capacity to **acquire and accumulate discriminative knowledge**.
>
> We sincerely appreciate your thoughtful feedback and hope the above clarifications are helpful. We welcome further discussion and are grateful for the opportunity to improve our work through your insights. Thank you again for your time and consideration.

---

> > ### Comment · Reviewer_nsjy · 2025-08-03
> >
> > Thanks for the clarification and additional evaluation. The authors' rebuttal has addressed my concerns, so I will increase my score to 4.  I recommend incorporating these experiments into the final version, as this would make the paper more comprehensive and persuasive.

---

> > > ### Author Response · Authors · 2025-08-04
> > >
> > > Dear Reviewer nsjy,
> > >
> > > Thank you sincerely for your time and effort in reviewing our paper. We are truly grateful for your positive recognition and the increased recommendation score. Your insightful feedback has been invaluable in enhancing the clarity and comprehensiveness of our work. We will carefully incorporate the suggested additional experiments and discussion into the revised manuscript.
> > >
> > > Best regards,
> > >
> > > The Authors

---

### Note · Authors · 2025-08-12

Dear NeurIPS Reviewers, ACs, and SPCs,

We sincerely thank the reviewers for their professional feedback, which has greatly helped improve our manuscript. We also appreciate the recognition of the **novelty**, **effectiveness**, and **good writing** of our work, as well as the **positive recommendations from all reviewers**.

During the review and rebuttal process, for **Reviewer #nsjy** (Rating: 3→4), we have addressed concerns regarding layout, typos, pretrained models, equations, adaptability to different scenarios, privacy leakage, and attention maps.

For **Reviewer #gSYz** (Rating: 4→4), we have addressed questions on the inclusion of additional datasets and the definition of the FAA metric.

For **Reviewer #nSnG** (Rating: 4→5), we have addressed points on the additional datasets and FAA metric definition.

For **Reviewer #Zi11** (Rating: 4→4), we have addressed concerns on improvements in class coherence, ablation of the KD loss, the mechanism of local class distribution compensation prompts, and typos.


In summary, our revised manuscript focuses on two main aspects:

(1) Correcting typos to improve clarity.

(2) Adding the additional experiment results and analysis to enhance comprehensiveness.


We again sincerely thank the Reviewers, ACs, and SPCs for the time and effort in evaluating our work.

Sincerely,

The authors

---

### Decision · Program_Chairs · 2025-09-17

**Decision:**

Accept (poster)

**Comment:**

This paper proposes C²Prompt, a novel method for Federated Continual Learning that addresses class-wise knowledge coherence via two key modules: Local Class Distribution Compensation (LCDC) and Class-aware Prompt Aggregation (CPA). The approach is well-motivated, clearly presented, and technically sound, with comprehensive experiments demonstrating consistent state-of-the-art performance across multiple datasets, model scales, and cross-dataset scenarios.

The authors’ rebuttal successfully addressed reviewers’ concerns, adding new experiments, clarifying metrics, and providing strong evidence of robustness under severe data heterogeneity. Given its novelty, solid empirical results, and practical relevance, this paper makes a meaningful contribution to the FCL community.